# Dorsal premammillary projection to periaqueductal gray controls escape vigor from innate and conditioned threats

Weisheng Wang[1†], Peter J Schuette[1†], Mimi Q La-Vu[1], Anita Torossian[2], Brooke C Tobias[1], Marta Ceko[3], Philip A Kragel[4], Fernando MCV Reis[1], Shiyu Ji[1], Megha Sehgal[5], Sandra Maesta-Pereira[2], Meghmik Chakerian[1], Alcino J Silva[1,5,6], Newton S Canteras[7], Tor Wager[4], Jonathan C Kao[8], Avishek Adhikari[1]*

[1]Department of Psychology, University of California, Los Angeles, Los Angeles, United States; [2]University of California, Los Angeles, Los Angeles, United States; [3]Institute of Cognitive Science, University of Colorado, Boulder, United States; [4]University of Colorado, Boulder, United States; [5]Department of Neurobiology, University of California, Los Angeles, Los Angeles, United States; [6]Department of Psychiatry & Biobehavioral Sciences, University of California, Los Angeles, Los Angeles, United States; [7]Department of Anatomy, University of São Paulo, Sao Paulo, Brazil; [8]Electrical and Computer Engineering, University of California, Los Angeles, Los Angeles, United States

**\*For correspondence:**
avi@psych.ucla.edu

[†]These authors contributed equally to this work

**Competing interest:** The authors declare that no competing interests exist.

**Abstract** Escape from threats has paramount importance for survival. However, it is unknown if a single circuit controls escape vigor from innate and conditioned threats. Cholecystokinin (cck)-expressing cells in the hypothalamic dorsal premammillary nucleus (PMd) are necessary for initiating escape from innate threats via a projection to the dorsolateral periaqueductal gray (dlPAG). We now show that in mice PMd-cck cells are activated during escape, but not other defensive behaviors. PMd-cck ensemble activity can also predict future escape. Furthermore, PMd inhibition decreases escape speed from both innate and conditioned threats. Inhibition of the PMd-cck projection to the dlPAG also decreased escape speed. Intriguingly, PMd-cck and dlPAG activity in mice showed higher mutual information during exposure to innate and conditioned threats. In parallel, human functional magnetic resonance imaging data show that a posterior hypothalamic-to-dlPAG pathway increased activity during exposure to aversive images, indicating that a similar pathway may possibly have a related role in humans. Our data identify the PMd-dlPAG circuit as a central node, controlling escape vigor elicited by both innate and conditioned threats.

## Introduction

In the presence of life-threatening danger, animals must quickly flee to minimize risk (*Perusini and Fanselow, 2015*). Due to the vital importance of escape for survival, the neural circuits controlling escape from threats have been extensively studied. The structure studied most commonly in escape is the dorsolateral periaqueductal gray (dlPAG). Stimulation of the dlPAG provokes rapid escape in rodents (*Deng et al., 2016*; *Evans et al., 2018*) and panic-related symptoms in humans (*Nashold et al., 1969*). Furthermore, single-unit dlPAG recordings show that a high proportion of cells are activated during escape (*Deng et al., 2016*; *Evans et al., 2018*). In agreement with these data, it has been shown that the dlPAG controls escape vigor, measured by escape velocity (*Evans et al., 2018*). However, inputs to the dlPAG that may control escape vigor have not been identified. The dorsomedial portion of the ventromedial hypothalamus (VMHdm) is a major excitatory dlPAG input,

suggesting that the VMHdm projection may mediate escape. However, activation of the VMHdm projection to the dlPAG surprisingly caused freezing, not escape (*Wang et al., 2015*). The other main hypothalamic input to the dlPAG is the dorsal premammillary nucleus (PMd) (*Canteras and Swanson, 1992*; *Tovote et al., 2016*). Surprisingly, despite being the strongest known input to the panicogenic dlPAG (*Canteras and Swanson, 1992*; *Tovote et al., 2016*), the activity of this nucleus has not been directly manipulated or recorded.

The PMd is a key component of the hypothalamic defense system and is strongly activated by various imminent threats (*Cezario et al., 2008*). Dangerous stimuli that activate the rodent PMd are extremely diverse and include carbon dioxide (*Johnson et al., 2011*), several predators (cats, snakes, and ferrets) (*Mendes-Gomes et al., 2020*), as well as aversive lights and noises (*Kim et al., 2017*). Additionally, the PMd is also activated by contexts fear-conditioned with shocks (*Canteras et al., 2008*) and social defeat (*Faturi et al., 2014*), indicating that it may play a role in coordinating defensive behaviors to both innate and conditioned threats. However, to date, the role of the PMd in escape vigor has not been directly studied. Furthermore, escape is generally studied during exposure to innate threats (*Deng et al., 2016*; *Evans et al., 2018*). Consequently, it is not known if escape from innate and conditioned threats requires the same circuit. Considering the PMd's involvement in innate and conditioned defense, as explained above, we predicted this region controlled escape from both threat modalities.

The vast majority of PMd cells are glutamatergic and express cholecystokinin (cck), and we recently showed that these cells controlled versatile context-specific escape from innate threats (*Wang et al., 2021*). Furthermore, inhibition of the PMd-cck projection to the dlPAG decreased the number of observed escapes induced by a range of innate threats, including a live predator and carbon dioxide. Conversely, the PMd-cck projection to the anteromedial thalamus (amv) was only recruited for escapes that required spatial navigation (*Wang et al., 2021*). However, it is unknown if PMd-cck cells also control escape velocity. Considering the results discussed above and prior reports showing the dlPAG controls escape vigor (*Evans et al., 2018*), we hypothesize that PMd-cck cells affect escape vigor via their projection to the dlPAG, but not the amv. We previously also showed that PMd-cck cells were active during escapes (*Wang et al., 2021*), but it remains unknown if these cells encode or predict future occurrences of escape and other defensive behaviors, and whether they represent relevant metrics such as distance to threat. Lastly, our prior study only investigated how PMd-cck cells affected escape caused by innate threats. It is unknown how this population is activated by conditioned threats and if PMd-cck cells affect defensive behaviors elicited by conditioned threats. To address these questions, we explored if PMd-cck cell activity is necessary for defensive behaviors elicited by innate and conditioned threats (a live predatory rat and a shock grid, respectively) (*Reis et al., 2021*). We also characterized how PMd-cck cells represent these threats and defensive behaviors during threat exposure.

Here, we show that PMd-cck cells encoded and predicted escape from innate and conditioned threats. Furthermore, inhibition of these cells or of their projection to the dlPAG decreased escape speed from a live predator or a conditioned threat (a shock grid). Lastly, functional magnetic resonance imaging (fMRI) data show that a hypothalamic-dlPAG pathway displays increased activation during exposure to aversive images, indicating that a similar pathway from a posterior medial hypothalamic nucleus to the brainstem may also exist in humans. These results show, for the first time, that the PMd is a vital node in coordinating escape from both innate and conditioned threats, and thus is likely to play key roles in minimizing exposure to danger.

## Results

### Innate and conditioned threats induce defensive behaviors

To study the PMd's role in controlling defensive behaviors, we exposed mice to two threats: a live predatory rat or a shock grid. These two assays were used to investigate, respectively, innate and conditioned threats. For the rat assay, mice were exposed either to a safe control toy rat or to an awake rat in a long box (70 cm length, 26 cm width, 44 cm height) for 10 min. The rat was placed at one of the corners, and its movement was restricted by a harness tied to a wall, restricting its range of motion to the rat area shown in pink in *Figure 1A*. Rats were screened for low aggression and predatory tendencies and thus they did not attack mice. No separating barrier was used between

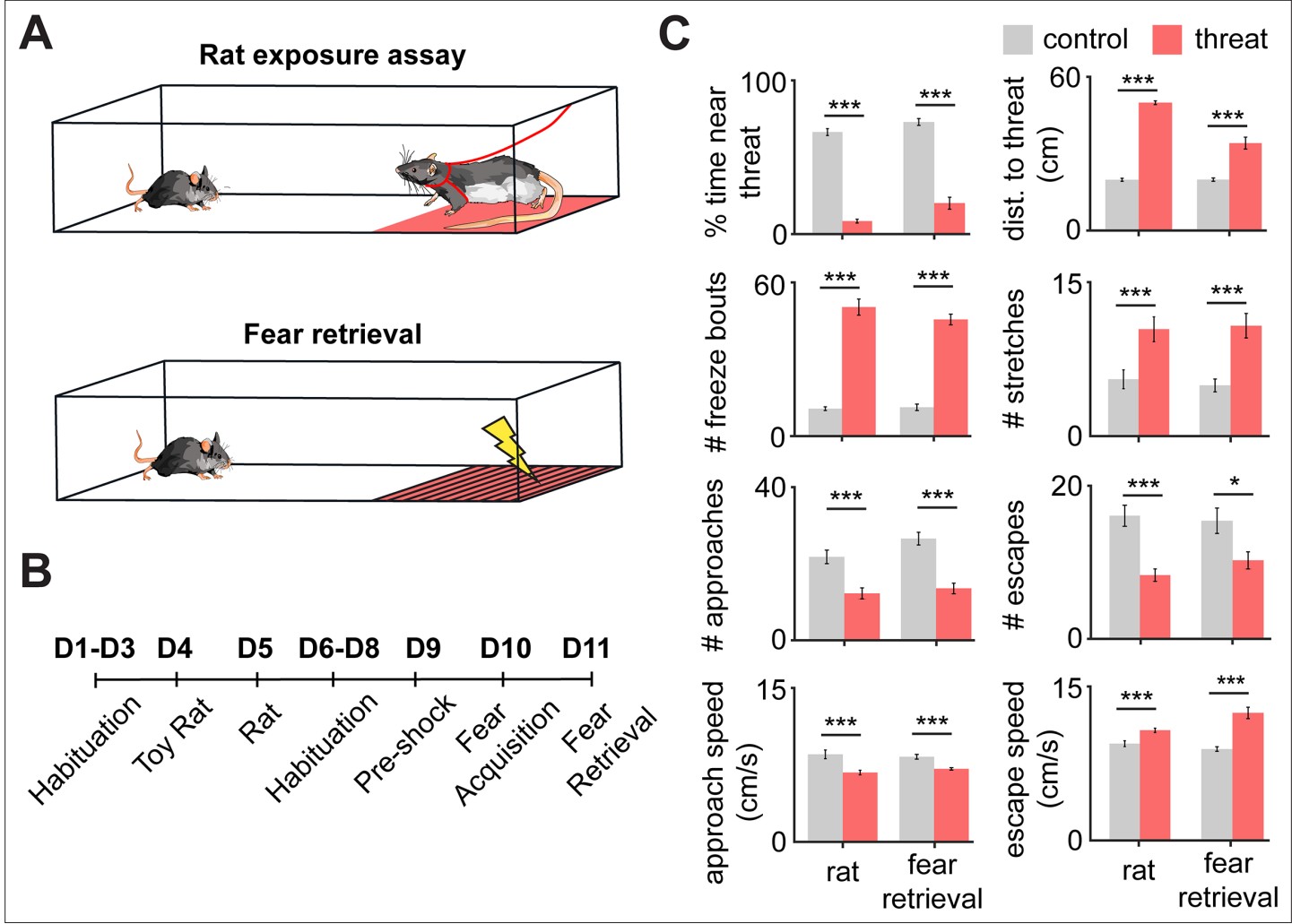

**Figure 1.** Rat and fear retrieval assays increased fear-related metrics. (**A**) Schemes of (top) rat assay and (bottom) fear retrieval assay. The rat is restricted by a harness (shown in red) that is tied to the upper wall edge and can only move in the pink area. In the shock grid assay, mice freely explored a context with a shock grid for three daily sessions (pre-shock, fear acquisition, and fear retrieval). Shocks were delivered only on fear acquisition day. All presented shock grid data is from fear retrieval. (**B**) Assays were performed in the order described (D = day). (**C**) Bars depict behavioral metrics (n = 32), for rat and fear retrieval assays, both for control and for threat conditions. Wilcoxon signed-rank test; *p<0.05, ***p<0.001.

The online version of this article includes the following figure supplement(s) for figure 1:

**Figure supplement 1.** Distribution of the difference scores for threat - control assays.

**Figure supplement 2.** The order of threat exposure does not affect defensive behavioral metrics.

**Figure supplement 3.** Distribution of the difference scores for threat–control assays for males and females.

rats and mice allowing for close naturalistic interactions. Rat and toy rat exposures were separated by 24 hr. For the shock grid assay, mice first explored a different box for three consecutive days for 10 min sessions. The shock grid was placed in one of the corners of the box, as shown in *Figure 1A*. On day 1, no shocks were given and mice freely explored the environment. On day two, a single 0.7 mA 2 s shock was given the first time the mouse touched the shock grid. On day 3 (fear retrieval), no shocks were given. All behavioral and neural data plotted from the shock grid is from the fear retrieval day, unless otherwise noted. The pre-shock baseline was used as the control for the fear retrieval day. All sessions were separated by 24 hr (*Figure 1B*). Threat exposure induced distance from the threat source, freezing, and stretch-attend postures (*Figure 1C*, *Figure 1—figure supplement 1*). (The mean freeze bout duration was 2.03 s ± 0.15.) Additionally, relative to control assays, during exposure to threat, approach velocity was lower, while escape velocity was higher (*Figure 1C*). These results

indicate that mice slowly and cautiously approach threats and then escape in high velocity back to safer locations far from threats.

We performed the rat exposure assay before the shock grid assay because the former is a milder experience than the latter; no actual pain is inflicted in the rat assay. We thus reasoned that the more intensely aversive assay (the shock assay) was more likely to influence behavior in the rat assay than vice-versa. Nevertheless, to determine whether there could be an effect of order, we exposed two cohorts of mice to the rat and shock grid threats in a counterbalanced manner and showed that behavior in the shock grid assay is not affected by prior experience in the rat assay (*Figure 1—figure supplement 2*). Taken together, these data show that both innate and conditioned threats induced defensive behaviors. Our data also support the view that escape velocity is a measure of threat-induced behavior. No sex differences were found in either behavioral assay (*Figure 1—figure supplement 3*) (male n = 17, female n = 15; Wilcoxon rank-sum test, p>0.05).

## PMd-cck cells are activated by proximity to threat and during escape

We next investigated the activity of PMd cells during threat exposure. To do so, we used a cck-cre line. We then injected AAV-FLEX-GCaMP6s in the PMd and implanted fiberoptic cannula above the injection site in cck-cre mice to record calcium transients in PMd-cck cells using fiber photometry (*Figure 2A–C*). Animals exhibited robust defensive behavior in the presence of threat (*Figure 2—figure supplement 1*). Examining the relationship of general locomotion and to the fiber photometry signal, we found that the signal amplitude was higher during threat exposure relative to control assays for a wide range of matched speed values (*Figure 2—figure supplement 2*). Averaged heat maps show PMd-cck activity was increased near the rat and the shock grid during fear retrieval (*Figure 2D*). Indeed, activity was increased near threats relative to control stimuli (toy rat and shock grid in pre-shock day). These comparisons were done when analyzing data at the same speed range (*Figure 2E*); thus, PMd-cck cells are more active near threats independently of locomotor changes. We next studied how PMd-cck cell activity changed during defensive behaviors. A representative trace suggests that these cells show high activity during escape (*Figure 2F*). Average data show that in both assays PMd-cck cells showed increased activation during risk-assessment stretch-attend postures and during escape, while a decrease in activity was displayed during freezing (*Figure 2G–I*). Furthermore, the total distance of each escape was correlated with PMd-cck activation during exposure to threats, but not control stimuli (*Figure 2J*). These results show that PMd-cck cells are quickly activated by proximity to threat and escape, during exposure to both innate and conditioned threats. In agreement with this view, PMd-cck cells displayed relatively high membrane input resistance (484 ± 64 MOhms) and low rheobase, which is the minimum current required to elicit an action potential (38.3 ± 6.1 pA) (*Figure 2—figure supplement 3*). These results indicate that fairly minor excitatory input is enough to activate these cells. These biophysical characteristics suggest that these cells may be rapidly activated in the presence of threats.

## PMd-cck ensemble activity predicts escape occurrence and flight vigor

To analyze how PMd-cck ensemble activity encodes escape, we implanted miniature head mounted fluorescent microscopes (miniscopes) above GCaMP6s-expressing PMd-cck cells (*Figure 3A,B*). Large ensembles of PMd-cck cells were recorded in the rat and shock grid assays (*Figure 3C,D*). Using a generalized linear model (GLM), we identified a large fraction of PMd-cck cells that are active during these behaviors (*Figure 3E*). The behavior that activated the largest and smallest number of PMd-cck cells was, respectively, escape and freezing (*Figure 3E*). These data agree with our fiber photometry results showing that bulk PMd-cck activity is highest during escape and lowest during freezing. Behavior-triggered averages indicate that PMd-cck cells may be significantly activated during defensive behaviors, in agreement with these results (*Figure 3F*). Further supporting a role for PMd-cck cells in escape, we show that ensemble activity could be used to decode ongoing escape, but not other behaviors (*Figure 3G*). These intriguing results raise the possibility that PMd-cck activity may be able to predict future occurrence of escape. Indeed, PMd-cck activity could predict escape from innate and conditioned threats several seconds prior to escape onset. However, ensemble activity could not predict movement away from control stimuli (toy rat and shock grid in pre-shock day) (*Figure 3H*). These data show that PMd-cck activity can specifically predict future escape from threats, but not moving away from objects in general. Additionally, we found that PMd-cck cells represent not only future escape onset (*Figure 3H*) but also escape speed. Using the correlation of single cell activity

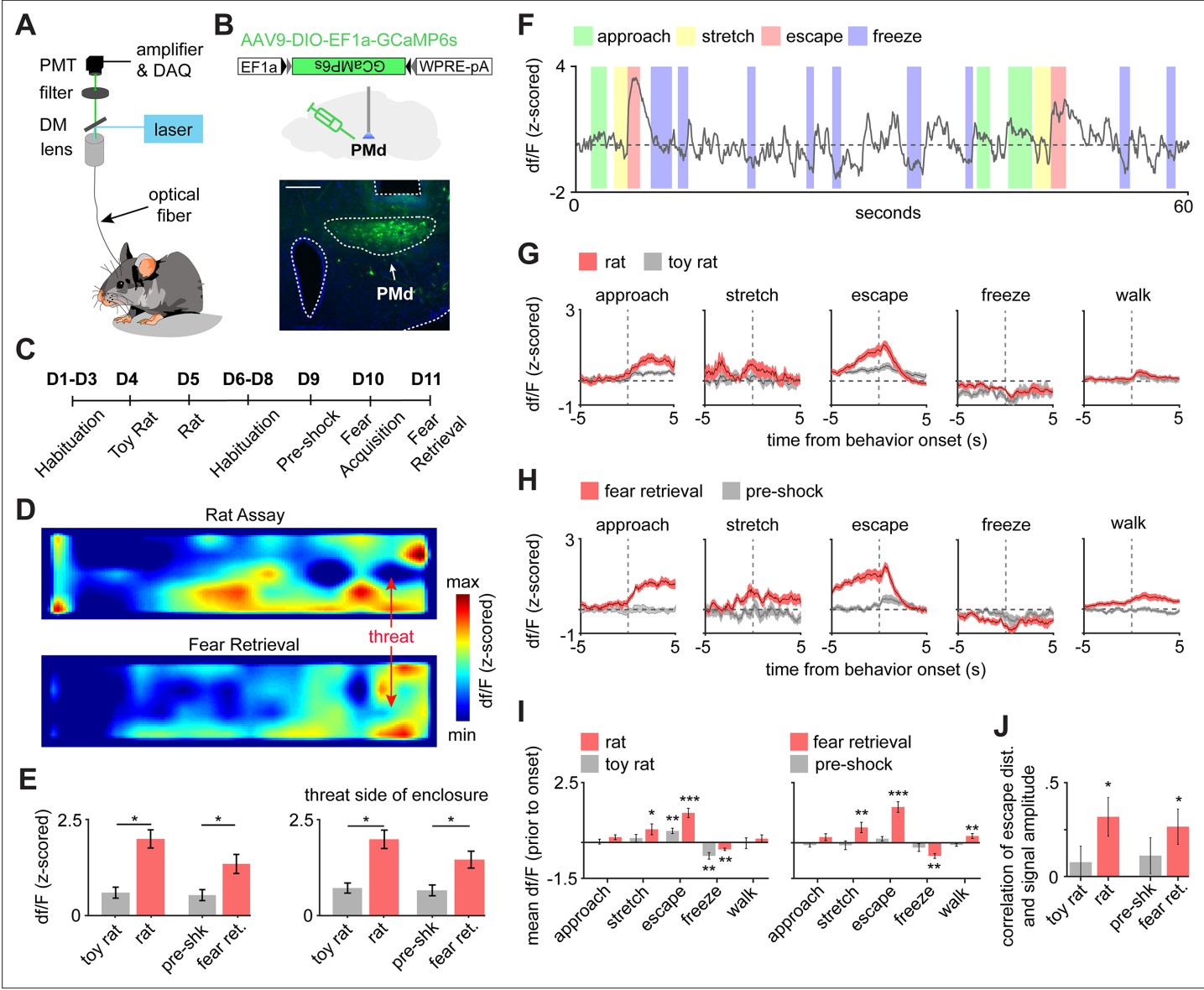

**Figure 2.** PMd-cck cells are activated by threats and escape. (**A**) Scheme showing setup used to obtain fiber photometry recordings. (**B**) Expression of GCaMP6s in PMd-cck cells. (scale bar: 200 μm). (**C**) Diagram depicts the behavioral protocol for each day (abbreviated as D). (**D**) Average heatmaps showing that PMd-cck cells are more active near a rat (top) and the shock grid (bottom) (for each, n = 15). (**E**) Bar graphs quantifying average z-scored df/F during exposure to the toy rat, rat, pre-shock, and fear retrieval. All data are shown for the same speed range (6–10 cm/s; Wilcoxon signed-rank test). (**F**) Example GCaMP6s trace from a representative mouse showing that PMd-cck cells are active during escape. (**G**) Behavior-triggered average showing mean PMd-cck activity during approach to rat, risk-assessment stretch-attend postures, escape, and freeze. (n = 15 mice) (**H**) Same as (**G**), but during exposure to the fear retrieval shock grid assay. (n = 15 mice). (**I**) Bars show the mean df/F from –2 to 0 s from behavior onset for threat (red) and control (gray) assays. (Wilcoxon signed-rank test; n [left] same as (**F**); n [right] same as (**G**)). (**I**) Bars show the Spearman correlation of the mean fiber photometry signal amplitude and distance run for all escapes. (Wilcoxon signed-rank test). (**E, I, J**), n = 15 mice, data is plotted as mean ± s.e.m. *p<0.05, **p<0.01, ***p<0.001.

The online version of this article includes the following figure supplement(s) for figure 2:

**Figure supplement 1.** Behavioral metrics for the PMd fiber photometry cohort during threat exposure assays.

**Figure supplement 2.** PMd-cck df/F for increasing speed and acceleration ranges.

**Figure supplement 3.** Characterization of *PMd-cck* cell biophysical properties in acute slices.

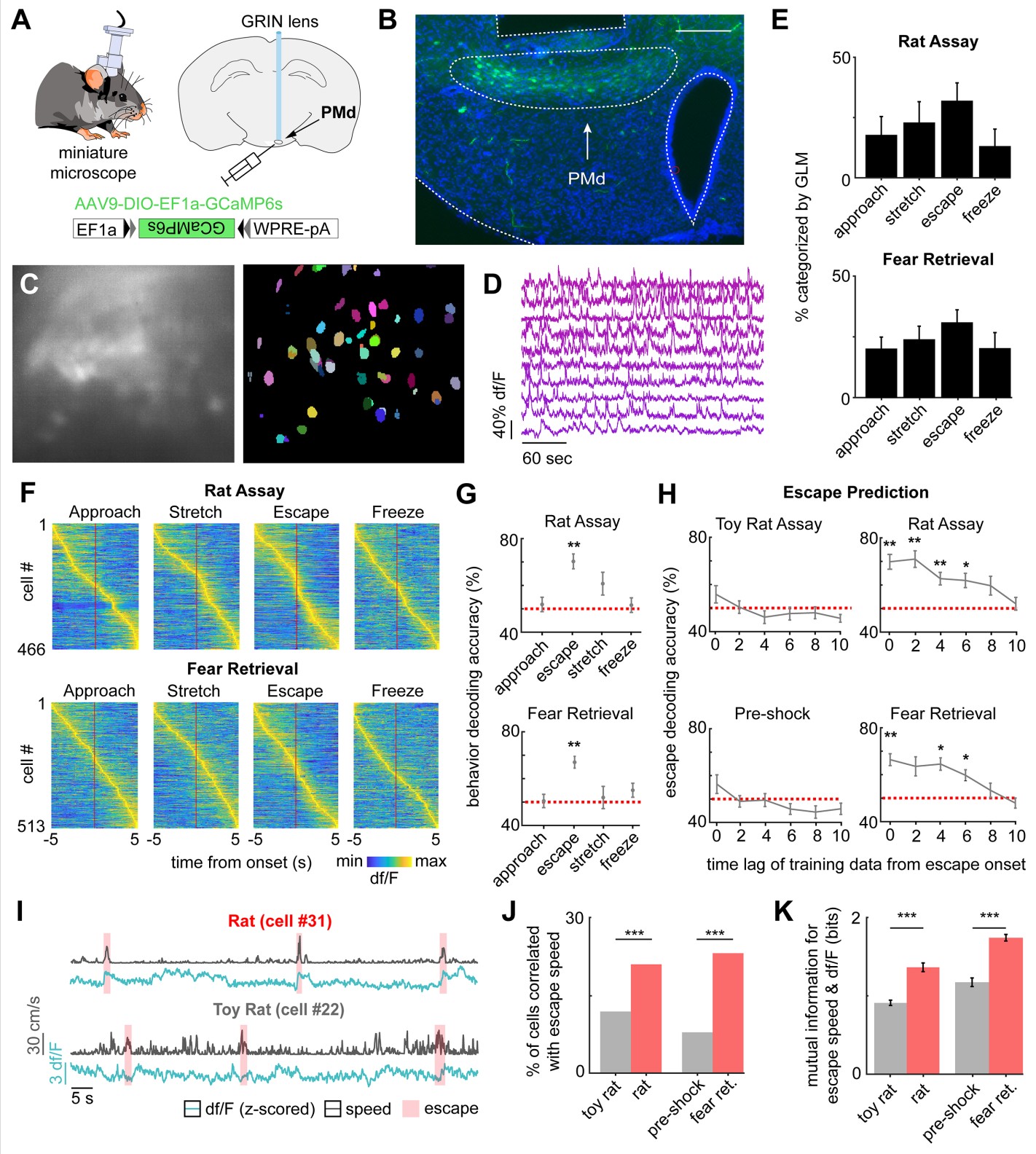

**Figure 3.** PMd-cck ensemble activity can predict escape in rat and shock grid fear retrieval assays. (**A**) PMd-cck mice were injected with AAV9-DIO-EF1a-GCaMP6s in the PMd and then were implanted with a miniaturized microscope. (**B**) Photograph of the GCaMP6s in PMd-cck cells and location of implanted GRIN lens. (Scale bar 200 μm) (**C**) (Left) Maximum projection of the PMd field of view in an example mouse. (Right) Extracted cell contours for the same session. (**D**) Representative traces of a subset of calcium transients from GCaMP6s-expressing PMd-cck cells recorded in a single session.

*Figure 3 continued on next page*

*Figure 3 continued*

(**E**) Generalized linear models (GLMs) were used to determine GLM weights for defensive behaviors. Cells were classified as activated by each behavior based on their actual GLM weights compared to the distribution of weights generated by permuting the neural data. (n = 9 mice) (**F**) Colormaps show average activation for each PMd-cck cell for each scored behavior in the rat (top) and shock grid fear retrieval (bottom) assays. Cells are sorted by time of peak activation. (**G**) Ongoing escape, but not other behaviors, can be decoded by PMd-cck cell activity in the rat (top) and shock grid fear retrieval assays (bottom) (Mice that displayed less than five instances of a given behavior were removed from the analysis: [top] approach n = 7, stretch n = 6, escape n = 7, freeze n = 6; [bottom] approach n = 5, stretch n = 4, escape n = 5, freeze n = 3; Wilcoxon signed-rank test.) (**H**) PMd-cck cell activity can predict escape from threats, but not control stimuli, several seconds prior to escape onset. (Toy rat n = 8 mice, rat n = 7, pre-shock n = 5, fear retrieval n = 5). (n = 466 cells in rat assay, n = 513 cells in shock grid fear retrieval assay; Wilcoxon signed-rank test) (**I**) Traces show the z-scored df/F (blue) and speed (gray) for one cell classified as a speed cell in the rat exposure assay (top) and one non-correlated cell from the toy rat assay (bottom). Individual escape epochs are indicated by red boxes. (**J**) Bars show the percentage of cells that significantly correlate with escape speed. (Fisher's exact test; toy rat: n correlated = 56, n non-correlated = 405; rat: n correlated = 100, n non-correlated = 366; pre-shock: n correlated = 50, n non-correlated = 571; fear retrieval: n correlated = 122, n non-correlated = 391) (**K**) Bars show the mutual information in bits between escape speed and calcium activity for cells whose signals were significantly correlated with escape speed in (**J**). (Wilcoxon rank-sum test; toy rat n = 56, rat n = 100; pre-shock n = 50, fear retrieval n = 122). \*\*\*p<0.001, \*\*p<0.01, \*p<0.05.

and escape speed, we classified escape speed cells (see Materials and methods) in the control and threat assays. A higher fraction of PMd cells showed activity significantly correlated with escape speed for threat than control stimuli (*Figure 3I*). Additionally, for these escape speed-correlated cells, the mutual information between escape speed and calcium signal is significantly greater during threat than control (*Figure 3K*). These data indicate that PMd-cck activity is related to defensive escape and speed in the presence of threat, rather than general locomotion.

Our fiber photometry results indicate that PMd-cck cells were more active during close proximity to threat (*Figure 2D*). These data suggest that PMd-cck ensemble activity may represent position in threat assays. We thus decoded position in both control and threat assays using PMd-cck ensemble activity. Strikingly, the error of position decoding was both smaller in threat than in control assays and significantly less than chance error (*Figure 4A–B*). These results show that PMd-cck cells represent distance to threat more prominently than distance to control objects.

Having observed that a greater proportion of PMd cells correlate with speed (*Figure 3I,J*), we then studied if ensemble activity could predict movement vigor, measured by velocity. Indeed, PMd-cck activity could be used to decode velocity during threat exposure with higher accuracy than during exposure to control stimuli (*Figure 4—figure supplement 1*). Furthermore, decoding of velocity in control assays was less accurate than in threat assays, for both the rat and the shock assays (*Figure 4—figure supplement 1*). Since PMd ensemble activity can predict future escape, but not approach, we hypothesized that PMd activity could be used to decode velocity away from threats more accurately than velocity toward threats. Representative traces showing predicted and observed velocity support this hypothesis (*Figure 4*). Indeed, averaged data across mice show that the error for predicted velocity is lower for decoding velocity away from threat compared to velocity toward threat. Moreover, only velocity away from threat can be predicted with an error significantly less than chance (*Figure 4C,D*). These data show that PMd-cck cells represent key kinematic variables related to rapid escape from threats.

## PMd-cck inhibition decreases escape vigor

Recordings of PMd-cck ensemble activity revealed that these cells are highly active during escape and that their activity can be used to decode escape (but not approach) velocity. Moreover, neural activity could only decode escape, but not other behaviors. We thus hypothesized that inhibition of PMd-cck cells would decrease escape velocity without affecting other defensive behaviors. To test this view, we expressed the inhibitory receptor hM4Di in PMd-cck cells (*Figure 5A*). We confirmed that the hM4Di receptor ligand clozapine-*N*-oxide (CNO) produced hyperpolarization (*Figure 5B*). We then exposed mice to the assays described in *Figure 1A*. Mice were exposed to each threat and control assay twice, following treatment with either saline or the hM4Di ligand CNO (*Figure 5C*). Inhibition of PMd-cck cells in CNO-treated mice decreased escape velocity from both threats, in line with our prediction (*Figure 5D*). Importantly, inhibiting these cells did not change velocity, while mice moved away from control safe stimuli (toy rat and shock grid prior to fear conditioning) (*Figure 5—figure supplement 1*). This manipulation did not change freezing or stretch-attend postures (*Figure 5D*), showing PMd-cck activity is selectively required for escape, rather than defensive behaviors in general.

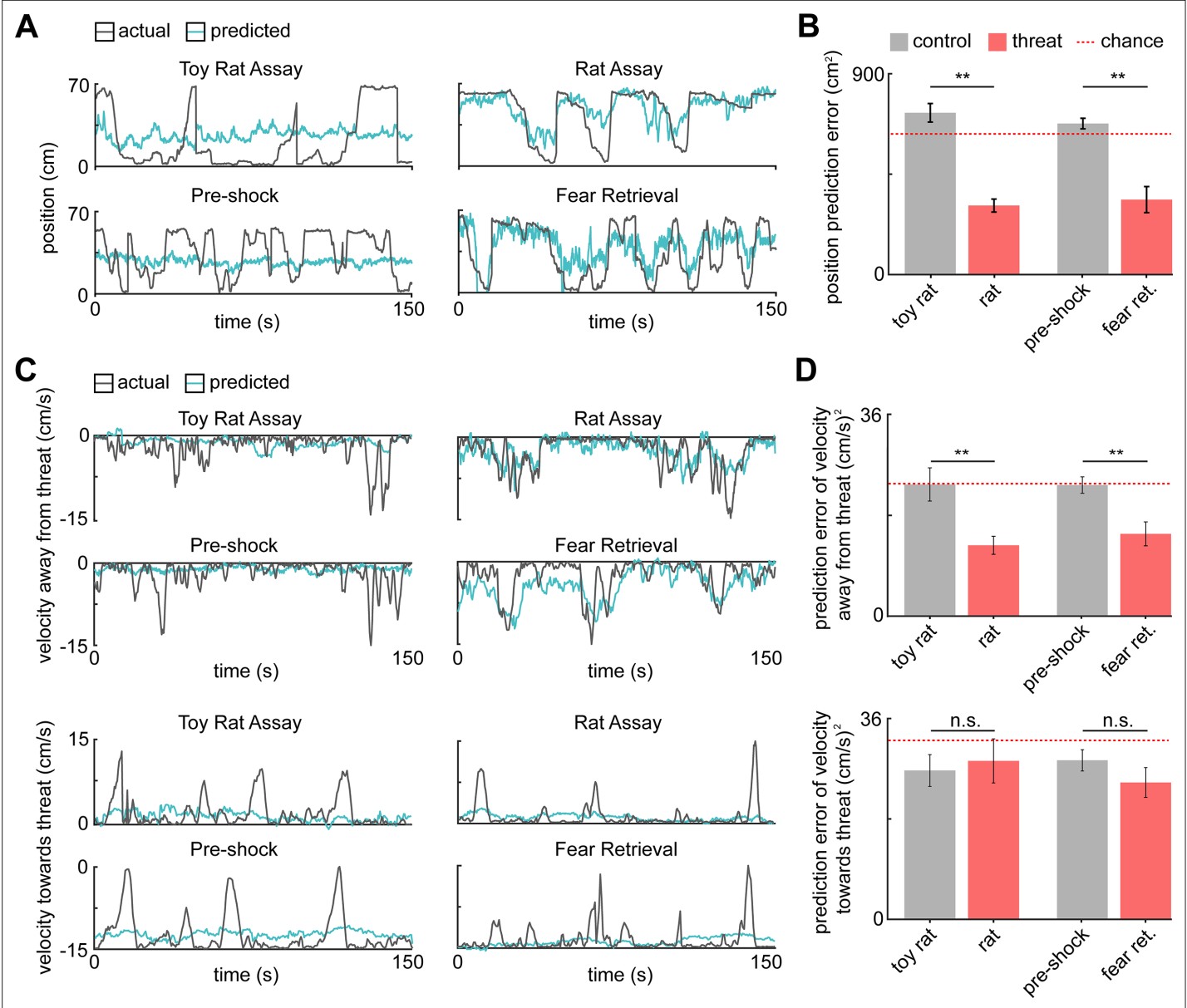

**Figure 4.** PMd ensemble activity represents distance from threat and escape velocity. (**A**) A general linearized model (GLM) was used to decode the position of each animal along the length of the enclosure from the neural data. The line plots depict the actual location (gray line) and GLM-predicted location (blue line) from example toy rat/rat and pre-shock/fear retrieval sessions. Note that the predicted location is more accurate for threat than control assays. (**B**) Bars show the mean squared error (MSE) of the GLM-predicted location from the actual location. The MSE is significantly lower for threat than control assays (Wilcoxon signed-rank test; n = 9 mice). The dotted red line indicates chance error, calculated by training and testing the GLM on circularly permuted data. Only threat assay error was significantly lower than chance (Wilcoxon signed-rank test; rat p<0.001, fear retrieval p=0.003). (**C**) Similar to (**A**), a GLM was used to predict the velocity away from (top) and toward (bottom) the threat in a representative mouse. (**D**) Similar to (**B**), bars depict the MSE of the GLM-predicted velocity away from (top) and toward (bottom) the threat. The GLM more accurately decodes threat than control velocities for samples in which the mice move away from the threat (top). As in (**B**), only threat assay error was significantly lower than chance (Wilcoxon signed-rank test; rat p=0.004, fear retrieval p=0.012). The accuracy does not differ in threat and control assays for samples in which the mice move toward the threat (bottom). (Wilcoxon test; n = 9 mice) **p<0.01.

The online version of this article includes the following figure supplement(s) for figure 4:

**Figure supplement 1.** PMd ensemble activity represents speed in threat assays.

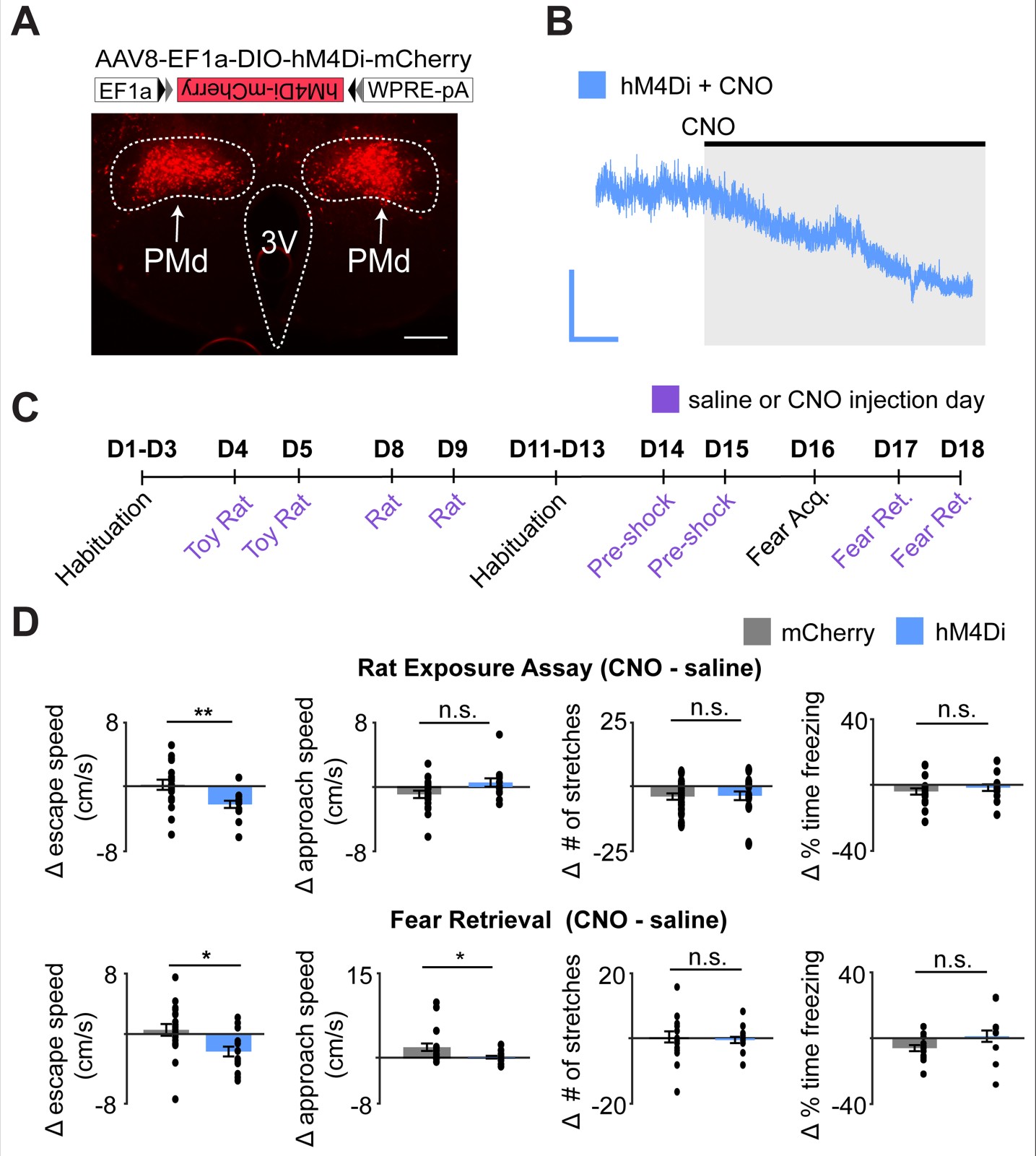

**Figure 5.** Chemogenetic inhibition of PMd-cck cells decreases escape speed from threats. (**A**) Cck-cre mice were injected with cre-dependent vectors encoding hM4Di-mcherry or -mcherry in the PMd (top). Expression of hM4Di-mcherry in PMd-cck cells (bottom). (scale bar: 200 μm) (**B**) Ex vivo slice recordings showing that clozapine-*N*-oxide (CNO) hyperpolarized PMd-cck cells expressing hM4Di (scale bar: 60 s, 10 mV). (**C**) Mice were exposed to each assay twice, in the order shown, after receiving i.p. injections of either saline or CNO. (**D**) Inhibition of hM4Di-expressing PMd-cck cells decreased

*Figure 5 continued on next page*

*Figure 5 continued*

escape speed in the rat and fear retrieval assays. (rat exposure assay mCherry/hM4Di n = 19/n = 11; fear retrieval assay mCherry/hM4Di n = 19/n = 12; Wilcoxon signed-rank test) **p<0.01, *p<0.05.

The online version of this article includes the following figure supplement(s) for figure 5:

**Figure supplement 1.** Inhibition of PMd-cck cells does not affect escape speed in control assays.

## Activation of PMd-cck cells recruits a wide network of regions involved in defensive behaviors

As PMd-cck inhibition decreases escape velocity, but not other behaviors, we predicted that activating these cells would specifically induce running and escape-related motion. Indeed, optogenetic activation of ChR2-expressing PMd-cck cells caused an increase in speed, but not in the amount of freezing or stretch-attend postures (*Figure 6A,B*).

We next investigated which downstream regions are recruited following activation of PMd-cck cells. Prior studies showed that the PMd projects to several structures involved in defense, such as the dlPAG and the anterior hypothalamus. Interestingly, it also projects to the anteromedial ventral thalamus (amv) (*Canteras and Swanson, 1992*). The amv has head direction cells (*Bassett et al., 2007*) and is a region critical for spatial navigation (*Jankowski et al., 2013*) and threat-conditioned contextual memory (*Carvalho-Netto et al., 2010*).

We hypothesized that activation of PMd-cck cells would recruit not only these known direct downstream areas, but also other structures involved in mounting a defensive behavioral state and regions involved in escape-related motor actions. To test this hypothesis, we optogenetically activated ChR2-expressing PMd-cck cells with blue light for 10 min (20 Hz, 5 ms pulses). Following perfusion, we performed an antibody stain against the immediate early gene cfos. PMd activation increased fos expression in regions that it projects to, such as the amv and the dlPAG. Interestingly, other nuclei critical for defensive behaviors, such as the basolateral amygdala, lateral septum, and the bed nucleus of the stria terminalis, were also activated (*Figure 6C*), even though they are not innervated by the PMd (*Canteras and Swanson, 1992*). These results show that the PMd recruits not only its direct downstream outputs, but also other regions involved in threat-related defense. Striatal regions were also activated, such as the caudate nucleus, possibly due to the hyperlocomotion and escape-related actions observed during optogenetic stimulation. Importantly, not all regions were engaged, showing functional specificity. For example, the dentate gyrus and the PMd-adjacent ventral PMd did not show increases in fos expression following PMd stimulation (*Figure 6D*). These data show that PMd-cck cells can recruit a broad network of threat-activated regions, which may contribute to a transition to a defensive state. Despite these intriguing data, it is possible that endogenous natural PMd activation does not result in recruitment of the same nuclei seen following optogenetic PMd-cck activation.

## The DlPAG is active during escape

To identify which PMd downstream targets control escape, we studied its two main outputs, the amv and the dlPAG (*Canteras and Swanson, 1992*). The amv has head direction cells (*Bassett et al., 2007*) and is a region critical for threat-conditioned contextual memory (*Carvalho-Netto et al., 2010*) and spatial navigation (*Jankowski et al., 2013*).

The amv is also necessary for the acquisition of contextual fear elicited by predators (*Carvalho-Netto et al., 2010*), demonstrating this region has a role in defensive behaviors. In contrast, the dlPAG is a critical node in the escape network (*Del-Ben and Graeff, 2009*; *Tovote et al., 2016*).

To identify which of these PMd outputs control escape speed, we injected AAV9-syn-GCaMP6s in wild-type mice in either the amv or the dlPAG and obtained calcium transient recordings in the rat and shock grid assays (*Figure 7A*). DLPAG activity increased during escape from the rat (*Figure 7C*), in agreement with prior work showing this region is active during escape from innate threats (*Deng et al., 2016*; *Evans et al., 2018*). However, the dlPAG also showed increased activity during exposure to the fear conditioned shock grid during fear retrieval (*Figure 7D,E*). To our knowledge, there are no prior reports showing the dlPAG is active during escape from conditioned threats. Surprisingly, like the dlPAG, the amv was also active during escape from both threat modalities (*Figure 7G–I*), even though there are no prior reports implicating the amv in escape.

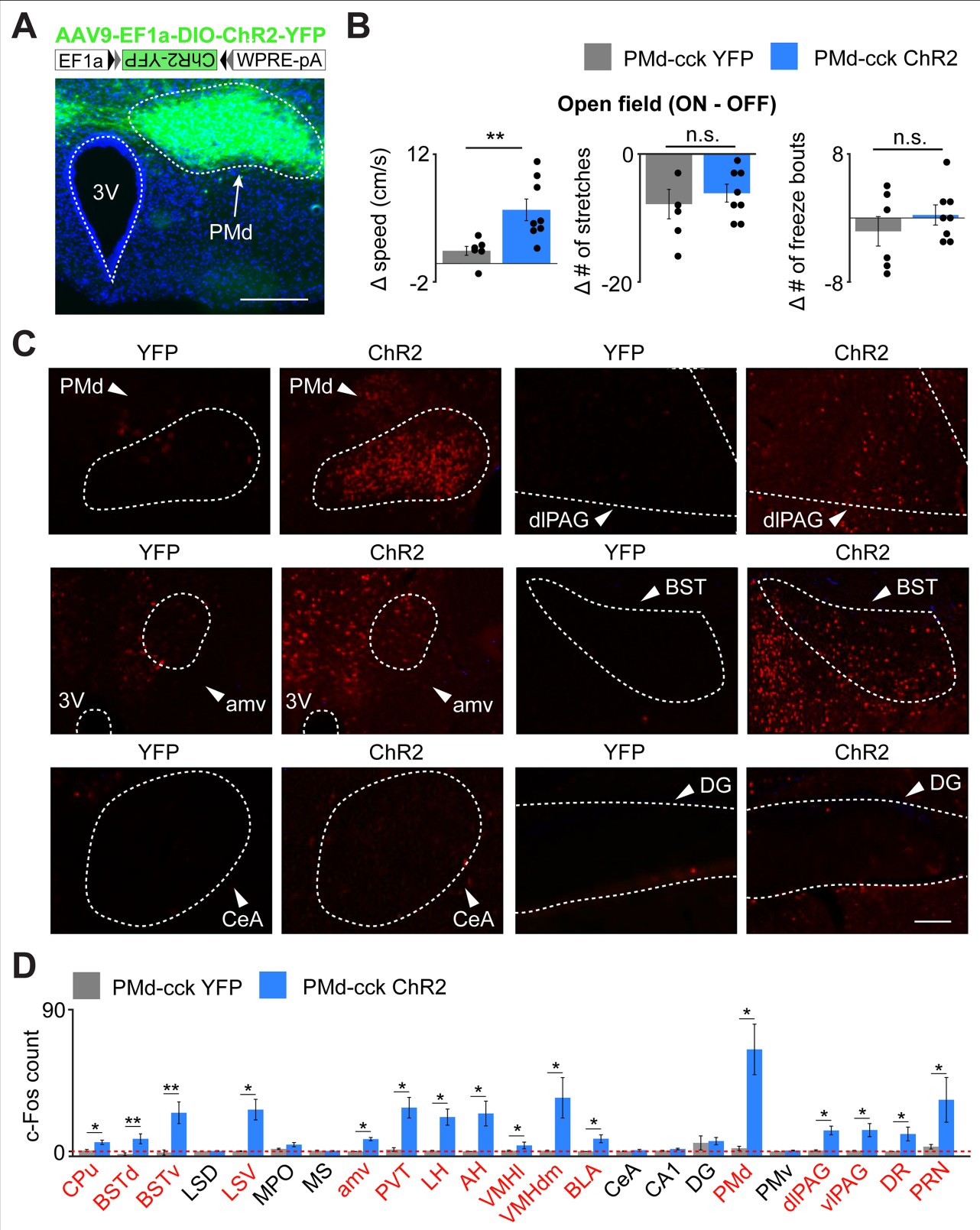

**Figure 6.** Optogenetic PMd-cck activation increases velocity and recruits widespread defensive networks. (**A**) Cck-cre mice were injected with AAV9-Ef1a-DIO-ChR2-YFP in the PMd (top). Expression of Chr2-YFP in PMd-cck cells (bottom; scale bar: 200 µm). (**B**) Delivery of blue light increases speed in PMd-cck ChR2 mice, but not stretch-attend postures or freeze bouts. (PMd-cck YFP n = 6, PMd-cck ChR2 n = 8; Wilcoxon rank-sum test). (**C**) Following optogenetic activation of PMd-cck cells, mice were perfused and stained with antibodies against the immediate early gene cfos. Representative

*Figure 6 continued on next page*

*Figure 6 continued*

images show that blue light delivery caused increased fos expression in the PMd, bed nucleus of the stria terminalis (BST) and anteromedial ventral thalamus (amv). Other regions, such as the central amygdala (Cea) and the dentate gyrus (DG) did not show increased fos expression following PMD-cck optogenetic stimulation. (scale bar: 100 μm) (**D**) Average number of fos-expressing cells in various brain regions following light delivery to ChR2 (blue) or YFP (gray)-expressing cells. Regions for which the c-Fos count is significantly greater for ChR2 than YFP mice are labeled in red. (Wilcoxon rank-sum test; for all regions, PMd-cck YFP n = 5, PMd-cck ChR2 n = 4 except for BSTd and BSTv: YFP n = 8, ChR2 n = 8) *p<0.05, **p<0.01. Abbreviations: CPu (caudate-putamen), BSTd/v (dorsal and ventral bed nucleus of the stria terminalis), LS D/V (dorsal and ventral lateral septum), MPO (medial preoptic area), amv (anteromedial ventral thalamus), PVT (paraventricular nucleus of the hypothalamus), LH (lateral hypothalamus), AH (anterior hypothalamus), VMHvl/dm (ventrolateral and dorsomedial portions of the ventromedial hypothalamus), BLA (basolateral amygdala), CeA (central amygdala), CA1 (hippocampal cornus ammonis 1), DG (dentate gyrus), PMd (dorsal premammillary nucleus), PMv (ventral premammillary nucleus), dlPAG (dorsolateral periaqueductal gray), vlPAG (ventrolateral periaqueductal gray), DR (dorsal Raphe), PRN (pontine reticular nucleus).

## Inhibition of the PMd-cck projection to the dlPAG decreases escape speed

Our fiber photometry results show that both major outputs of the PMd to the dlPAG and the amv are active during escape from threats, indicating the PMd-cck projections to these regions may control escape vigor. To identify which projection controls escape vigor, we expressed the inhibitory opsin Arch in PMd-cck cells and implanted fiberoptic cannulae bilaterally over either the amv or the dlPAG (*Figure 8A–C*). Inhibition of the PMd-cck projection to the dlPAG with green light decreased escape velocity in both assays (*Figure 8D*). This manipulation did not alter other defensive behaviors such as freezing or stretch-attend postures (*Figure 8D*). In contrast, inhibition of the PMd-cck projection to the amv did not change any defensive behavioral measure in either assay (*Figure 8E*). These data show that the activity in the PMd-cck projection to the dlPAG, but not to the amv, is necessary for normal escape vigor during exposure to both innate and conditioned threats.

## PMd and dlPAG show increased mutual information during threat exposure

Having shown that inhibition of the PMd-dlPAG projection impairs escape from threat, we hypothesized these regions show increased functional connectivity during threat exposure. To test this view, using cck-cre mice, we injected AAV-dio-GCaMP6s in the PMd and AAV-syn-GCaMP6s in the dlPAG contralaterally and implanted fiber optic cannula above each injection site to monitor the simultaneous calcium activity of these regions during threat and control assays (*Figure 9A–C*). Using the mutual information metric – an information-theoretic quantity that reflects the amount of information obtained for one variable by observing another variable – we found that the mutual information between the PMd and dlPAG is higher during exposure to threat than control assays (see Materials and methods for details). This was also true when escapes were removed, indicating that the mutual information change seen is related to threat exposure, rather than specific defensive behaviors (*Figure 9D*).

We opted to use mutual information instead of correlation because the former, but not the latter, can quantify both linear and non-linear relationships between two variables. Importantly, these dual-site recordings were done in PMd-cck cells and dlPAG-syn cells contralaterally.

As the PMd-cck projection to the dlPAG is unilateral (*Figure 9—figure supplement 1*; *Canteras and Swanson, 1992*), performing contralateral recordings ensures that dlPAG-syn cell body signals are not contaminated by signals from GCaMP-expressing PMd-cck axons terminating in the dlPAG. The dlPAG does not project to the PMd (*Comoli et al., 2000*); thus, there is no risk of recording signals from GCaMP-expressing dlPAG axons in the PMd.

## Hypothalamic-PAG functional connectivity increases in humans viewing aversive images

To investigate whether a functionally similar pathway exists in humans, we examined functional connectivity (i.e., covariation of BOLD signal in the hypothalamus and PAG) as participants received aversive stimulation during fMRI scanning (N = 48). We developed a predictive model to identify a pathway between the hypothalamus (HTH) and the PAG, which consisted of a multi-voxel pattern across brain voxels in each region optimized for maximal HTH–PAG covariation (*Figure 10*, see Materials and methods). We then tested activation in this pathway in held-out participants. This HTH–PAG

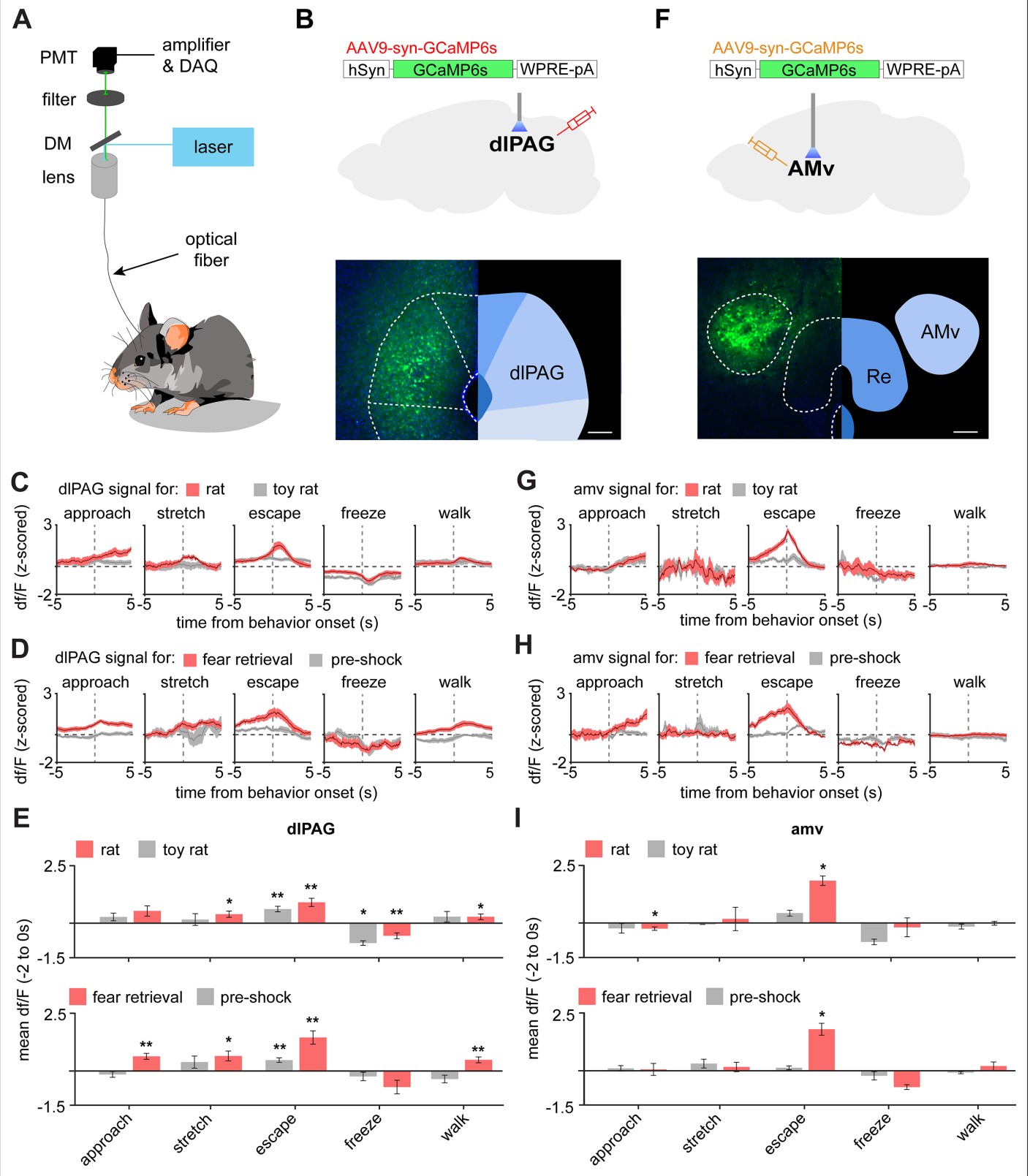

**Figure 7.** The dlPAG and AMV are activated by threats and escape. (**A**) Scheme showing setup used to obtain fiber photometry recordings. (**B**) Expression of GCaMP6s in the dlPAG. (Scale bar: 150 μm) (**C**) Behavior-triggered average showing mean dlPAG activity during approach to rat, risk-assessment stretch-attend postures, escape, freeze, and walking perpendicularly to the rat at the safe side of the enclosure. (n = 9 mice) (**D**) Same as (**C**), but during exposure to the fear retrieval shock grid assay. (n = 9 mice) (**E**) Bars show the mean df/F from –2 to 0 s from behavior onset for threat

*Figure 7 continued on next page*

Figure 7 continued

(red) and control (gray) assays. (n = 9 mice). (**F–I**) Same as (**B–E**), but for the amv. (**F**) Scale bar: 150 μm (**G–I**) n = 6 mice. (**E,F**) Wilcoxon signed-rank test; **p<0.01, *p<0.05.

pathway responded more strongly to aversive images than non-aversive images and its activation also scaled monotonically with aversiveness (*Figure 10C*). Examination of the multi-voxel patterns contributing to the HTH–PAG pathway revealed that portions of medial posterior hypothalamus (neighboring the mammillary bodies) were most consistently associated with PAG activation. We also show that activation of the HTH–PAG pathway is selective and does not correlate with activation of a different major subcortical input pathway to the PAG, such as the amygdala–PAG pathway (*Figure 10—figure supplement 1*).

Thus, these data show that functional connectivity in a hypothalamic-PAG pathway is increased in humans during aversive situations, in agreement with our results in mice showing that the hypothalamic to brainstem PMd–dlPAG pathway is engaged during exposure to aversive threats (*Figure 9*).

## Discussion

The PMd is anatomically the source of the most prominent input to the dlPAG (*Del-Ben and Graeff, 2009*; *Lovick, 2000*; *Tovote et al., 2016*). A wealth of evidence from diverse streams of data have demonstrated that the dlPAG controls escape (*Del-Ben and Graeff, 2009*; *Tovote et al., 2016*). Recent work has also shown that the dlPAG controls escape vigor (*Evans et al., 2018*). Taken together, these data indicate that the PMd is anatomically well-situated to modulate escape vigor from threats. Furthermore, optogenetic activation of PMd-cck cells activates a broad network of regions involved in defensive behaviors (*Figure 6D*). Our fos data show PMd-cck cell optogenetic activation recruited a plethora of areas known to mediate defense, such as the basolateral amygdala, the lateral septum and the bed nucleus of the stria terminalis. These results indicate that PMd activation potentially may affect a wide range of defensive behaviors by engaging these networks.

Our previous data showed that activation of the PMd–dlPAG pathway caused escape from innate threats (*Wang et al., 2021*). However, those data did not show if this circuit controlled escape vigor (measured by flight velocity) or if it affected escapes from conditioned threats. We now show PMd-cck cells play a key role in controlling escape vigor, during exposure to both innate and conditioned threats. We show that PMd-cck cells were activated by threat proximity (*Figures 2E and 4B*) and that their activity predicted future escape (*Figure 3H*) and represented escape velocity, but not approach velocity (*Figure 4D*). Furthermore, inhibition of either PMd-cck cells (*Figure 5*) and of the PMd-cck to dlPAG inhibition decreased escape velocity (*Figure 7*). These data demonstrate the PMd-cck projection to the dlPAG is critical for modulating escape velocity from threats, which is a behavior of paramount importance for survival. Importantly, all of the results described above are novel and were not shown in prior reports about the PMd (*Wang et al., 2021*).

Interestingly, PMd-cck cells also represented distance to threat, but not distance to control stimuli (*Figure 4*). PMd input to the dlPAG may thus contribute to the encoding of distance to threat and related kinematic variables in dlPAG cells as we recently reported (*Reis et al., 2021*; *Reis et al., 2021*). Prior work using excitotoxic PMd lesions and local infusions of muscimol in rats reported large decreases in freezing (*Cezario et al., 2008*). In contrast, our chemogenetic inhibition of PMd-cck cells in mice revealed only deficits in escape. These differences may be either due to differences in species or due to off-target effects of muscimol infusions in adjacent nuclei that control freezing, such as the VMHdm (*Wang et al., 2015*). Our data add to a growing stream of results showing how different components of the medial hypothalamic defense system control threat-induced behaviors, in a densely interconnected network containing the anterior hypothalamus, the VMHdm, and the PMd (*Cezario et al., 2008*).

Interestingly, our data show that the PMd, as well as the dlPAG, participate in defensive responses elicited by both innate and shock-based conditioned threats. The dlPAG has mostly been studied as a region that initiates escape from innate threats, such as looming stimuli (*Evans et al., 2018*). However, prior evidence has also implicated the dlPAG in conditioned defensive behavior. For example, the dlPAG is activated during exposure to shock-conditioned auditory tones and contexts (*Carrive et al., 1997*; *Watson et al., 2016*). Furthermore, neurotransmission of cannabinoids (*Resstel et al., 2008*),

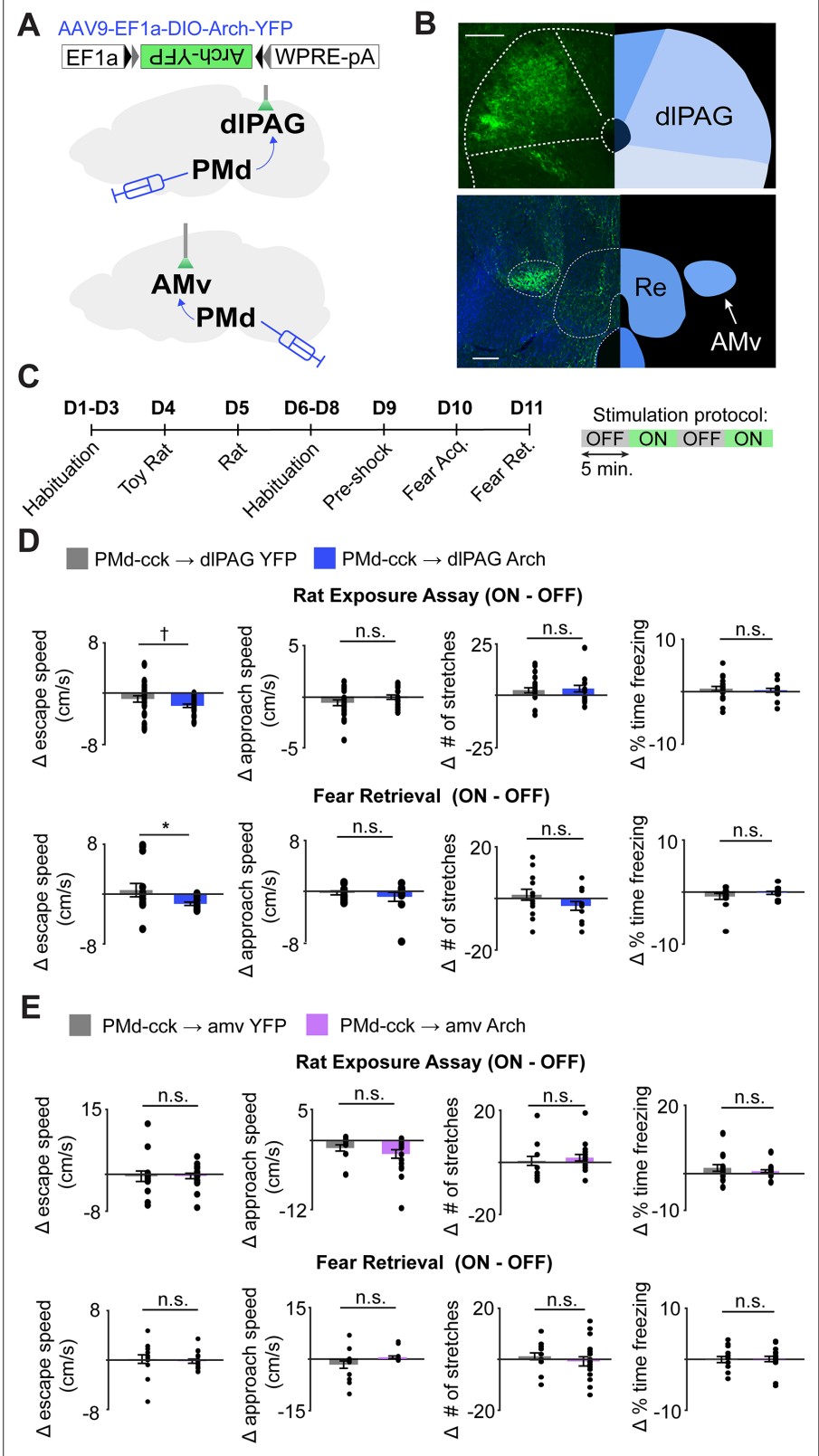

**Figure 8.** Optogenetic inhibition of the PMd-cck projection to the dlPAG, not the amv, decreases escape velocity during exposure to innate and conditioned threats. (**A**) Viral vectors were used to express Arch in PMd-cck cells. Fiber optic cannula were bilaterally implanted over PMd-cck arch-expressing axon terminals in the amv or dlPAG. (**B**) Image showing PMd-cck axon terminals expressing arch-YFP in the dlPAG and amv. (Scale bars: 150 µm)

*Figure 8 continued on next page*

*Figure 8 continued*

(**C**) Summary diagram showing order of assays and green light delivery protocol. (**D**) Inhibition of the PMd-cck projection to the dlPAG decreased escape speed, but not other defensive behaviors. (Wilcoxon rank-sum test; (top) rat exposure assay: YFP/Arch n = 24/n = 12; (bottom) fear retrieval: YFP/Arch n = 14/n = 11) (**E**) Inhibition of the PMd-cck projection to the amv did not alter any of the behavioral measures monitored. (Wilcoxon rank-sum test; (top) rat exposure assay: YFP/Arch n = 12/n = 18; (bottom) fear retrieval: YFP/Arch n = 12/n = 17), *p<0.05; †p=0.058.

CRF (*Borelli et al., 2013*), glutamate, and nitric oxide (*Aguiar et al., 2014*) have been shown to be necessary for contextual freezing. However, involvement of the dlPAG or the PMd in controlling escape behavior from conditioned stimuli such as shock grids is less well-understood.

We now show that the PMd-cck projection to the dlPAG modulates escape velocity from conditioned threats, broadening the role of this circuit to include escape from learned threats. More recently, we showed that dlPAG cells represent distance from a learned conditioned threatening shock grid during fear retrieval, further supporting a role for this region in mediating defense induced by conditioned threats (*Reis et al., 2021*). The dlPAG is bidirectionally connected with diverse forebrain regions (*Motta et al., 2017*), while the PMd receives strong input from the medial prefrontal cortex (*Comoli et al., 2000*), which may explain how these regions respond to conditioned threats. Intriguingly, during contextual fear retrieval tests, rats showed increased PMd fos expression if they had free access to the conditioning chamber, but not if they were confined to this chamber (*Viellard et al.,*

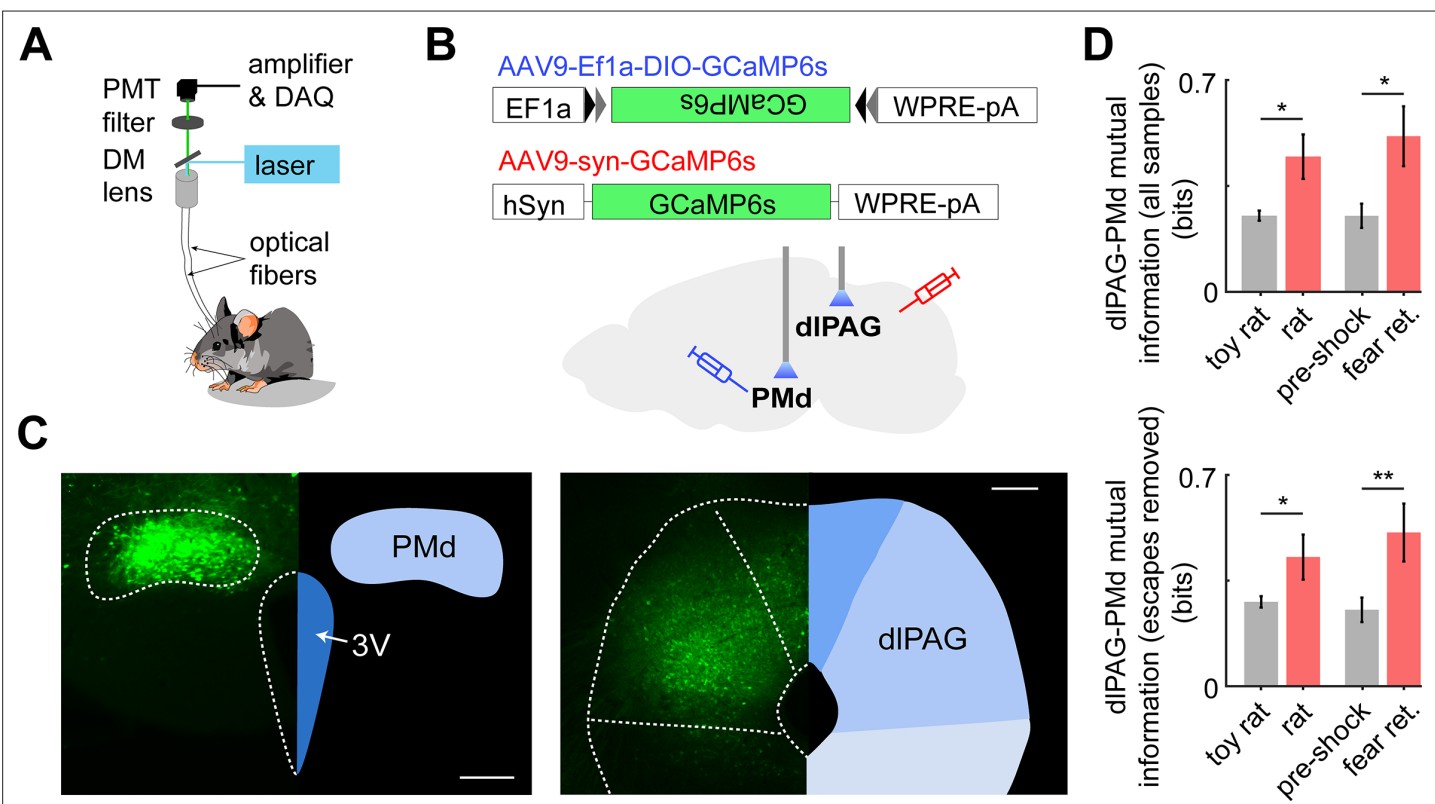

**Figure 9.** Dual fiber photometry signals from the PMd and dlPAG exhibit increased mutual information during threat exposure. (**A**) Scheme showing setup used to obtain dual fiber photometry recordings. (**B**) PMd-cck mice were injected with AAV9-Ef1a-DIO-GCaMP6s in the PMd and AAV9-syn-GCaMP6s in the dlPAG. (**C**) Expression of GCaMP6s in the PMd and dlPAG. (Scale bars: [left] 200 µm, [right] 150 µm) (**D**) Bars show the mutual information between the dual-recorded PMd and dlPAG signals, both including (left) and excluding (right) escape epochs, during exposure to threat and control. Mutual information is an information theory-derived metric denoting the amount of information obtained for one variable by observing another variable. See Materials and methods section for more details. *p<0.05, **p<0.01.

The online version of this article includes the following figure supplement(s) for figure 9:

**Figure supplement 1.** PMd-cck neurons project unilaterally to the dlPAG.

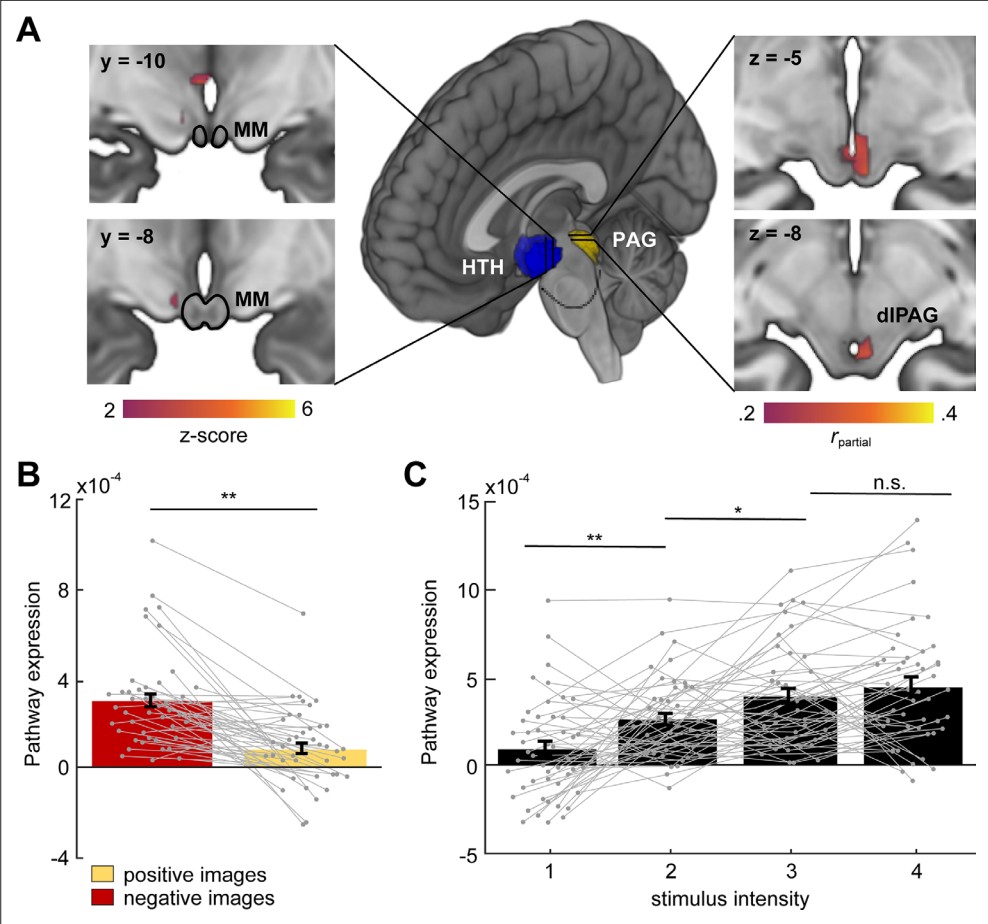

**Figure 10.** Hypothalamus (HTH)–PAG pathway is sensitive to aversive visual stimuli in humans. (**A**) Multivariate brain pathway estimated using activation in the hypothalamus (HTH, rendered in blue) to predict patterns of activation in the periaqueductal gray (PAG, rendered in yellow). Inserts depict statistical maps indicating which regions of the HTH covaried most strongly with the PAG (left) and portions of dorsal PAG (dlPAG) that were explained by the HTH but not a pathway from the central amygdala. The mammillary bodies (MM) are depicted with a black outline. Note that all hypothalamus voxels are included in the model, only suprathreshold voxels are shown here. (**B**) Average bar plot showing that the HTH–PAG pathway was more active during exposure to threat (aversive visual images) compared to control stimuli (non-aversive, positive images). Each circle corresponds to an individual subject. (**C**) Pathway expression monotonically increased as a function of stimulus intensity. Inference on brain maps is based on bootstrap resampling of regression coefficients from pathway estimation (left) and partial correlation coefficients (right). All maps are thresholded at $q_{FDR} < 0.05$. (Wilcoxon signed-rank test, n = 48 participants) **p<0.001, *p<0.01.

The online version of this article includes the following figure supplement(s) for figure 10:

**Figure supplement 1.** Multi-voxel response patterns in the PAG related to hypothalamus (HTH) and central amygdala (CeA) are functionally distinct.

*2016*). Information about innate predatory threats are likely conveyed to the PMd by other members of the hypothalamic predatory defense circuit, such as the VMHdm and the anterior hypothalamus (*Cezario et al., 2008*; *Comoli et al., 2000*; *Silva et al., 2013*). Future studies are needed to determine which specific inputs to the PMd convey information about conditioned threats. Nevertheless, our data show that the PMd–dlPAG circuit is not merely responding to external threatening sensory cues. Rather, the involvement of this circuit in escape from conditioned stimuli during fear retrieval shows that these structures can be affected by long-term fear memories, illustrating that evolutionarily ancient structures can also display experience-dependent roles in behavior.

Intriguingly, our prior data showed that optogenetic inhibition of the PMd–amv projection decreased the number of escapes elicited by a predator rat in environments requiring sophisticated

three-dimensional spatial navigation to escape. However, PMd–amv activity was not necessary for stereotyped jumps in the presence of the panicogenic agent $CO_2$ (**Wang et al., 2021**). One interpretation of these data is that this pathway is necessary only for escape from medium intensity threat modalities (such as a rat), but not from extremely high imminence threat such as $CO_2$. A second interpretation is that the PMd–amv pathway is only necessary for escapes that require spatial navigation, regardless of the threat modality. We now show inhibition of the PMd-cck projection to the amv did not alter any defensive behavioral metrics induced by a rat in a simple environment (**Figure 1**), where the animal does not need a complex three-dimensional understanding of the environmental layout to escape (**Figure 8D**). In the current assay, simply running away from the rat in any direction is sufficient to escape. As inhibition of the PMd–amv projection impaired escape from a predatory rat only when flight required complex navigation, we argue that the role of this circuit is related to complex navigation during threat exposure, supporting our second interpretation above. These data agree with prior work that indicate the amv's role in defensive behavior is related to contextual memory-associated behaviors rather than the execution of escape or freezing (**Carvalho-Netto et al., 2010**).

Intriguingly, our fMRI data indicate that a hypothalamic-PAG pathway has increased activity in humans viewing aversive images (**Figure 10**). A homologous functional pathway to the rodent PMd-dlPAG may exist in humans that is at least partially identifiable from fMRI data. We used a novel application of partial least squares (PLS) to identify local multi-voxel patterns that functionally connected HTH and dlPAG. In out-of-sample tests in new participants, HTH and dlPAG were positively correlated in every participant and tracked the reported intensity of negative emotion elicited by images. The resolution of imaging in humans does not allow us to specify which hypothalamic nucleus is involved. However, the location of the nucleus is in the posterior medial hypothalamus, similar to the rodent PMd, suggesting the possibility that a circuit analogous to the PMd–dlPAG projection may exist in humans. Despite these limitations and the differences between the tasks in human and rodent subjects, these data are compatible with rodent data showing the PMd is activated by a wide variety of aversive stimuli such as bright lights and loud noises (**Kim et al., 2017**). Furthermore, the fMRI data agree with our data showing in mice PMd-cck and dlPAG activity show increased mutual information in the presence of threat, relative to control conditions. The increase in PMd–dlPAG mutual information was present even after removing all samples with escape (**Figure 9**), indicating that this effect is related to exposure to threat, rather than being associated specifically with escape.

Taken together, our data indicate that the PMd-cck projection to the dlPAG modulates escape velocity during exposure to both innate and conditioned threats, and the results suggest a similar pathway may be active during exposure to aversive situations in humans.

## Materials and methods

All procedures conformed to guidelines established by the National Institutes of Health and have been approved by the University of California, Los Angeles Institutional Animal Care and Use Committee, protocols 2017–011 and 2017–075.

### Mice

Cck-IRES-Cre mice (Jackson Laboratory stock No. 012706) and wild-type C57BL/6 J mice (Jackson Laboratory stock No. 000664) were used for all experiments. Male and female mice between 2 and 6 months of age were used in all experiments. Mice were maintained on a 12 hr reverse light–dark cycle with food and water ad libitum. Sample sizes were chosen based on previous behavioral optogenetics studies on defensive behaviors, which typically use 6–15 mice per group. All mice were handled for a minimum of 5 days prior to any behavioral task.

### Rats

Male Long-Evans rats (250–400 g) were obtained from Charles River Laboratories and were individually housed on a standard 12 hr light–dark cycle and given food and water ad libitum. Rats were only used as a predatory stimulus. Rats were handled for several weeks prior to being used and were screened for low aggression to avoid attacks on mice. No attacks on mice were observed in this experiment.

## Viral vectors

All vectors were purchased from Addgene.

## Optogenetics

AAV9.EF1a.DIO.hChR2(H134R)-eYFP.WPRE.hGH, AAV9-EF1a-DIO-eYFP and AAV9-Ef1a-DIO-Arch-GFP. Chemogenetics: pAAV8-hSyn-DIO-hM4D(Gi)-mCherry and AAV8.Syn.DIO. mCherry
Fiber Photometry AAV9.Syn.GCaMP6s.WPRE.SV40 and AAV9.Syn.FLEX.GCaMP6s.WPRE.SV40

## Surgeries

Surgeries were performed as described previously (*Adhikari et al., 2015*). Eight-week-old mice were anaesthetized with 1.5–3.0% isoflurane and placed in a stereotaxic apparatus (Kopf Instruments). A scalpel was used to open an incision along the midline to expose the skull. After performing a craniotomy, 40 nl of one of the viral vectors listed above at a titer of $2 \times 10^{12}$ particles/ml was injected per site (PMd, amv, dlPAG) using a 10 µl nanofil syringe (World Precision Instruments) at 0.08 µl/min. AAV8-hSyn-DIO-hM4D(Gi)-mCherry and AAV8-hSyn-DIO-mCherry were injected at a titer of $2 \times 10^{12}$ particles/ml. The syringe was coupled to a 33-gauge beveled needle, and the bevel was placed to face the anterior side of the animal. The syringe was slowly retracted 20 min after the start of the infusion. Mice received unilateral viral infusion and fiber optic cannula implantation. Infusion locations measured as anterior-posterior, medial-lateral, and dorso-ventral coordinates from bregma were as follows: dlPAG (−4.75,–0.45, −1.9), dorsal PMd (−2.46,–0.5, −5.35), and amv (−0.85,–0.5, −3.9). For optogenetic experiments, fiber optic cannula (0.22 NA, 200 µm diameter; Newdoon) were implanted bilaterally 0.15 mm above the viral infusion sites. Only mice with viral expression restricted to the intended targets were used for behavioral assays.

For photometry experiments, mice were injected with 0.16 µl at a titer of $3 \times 10^{12}$ of AAV9.Syn.Flex. GCaMP6s.WPRE.SV40 in the PMd of cck-cre mice. The same volume and titer of AAV9.Syn.GCaMP6s. WPRE.SV40 was injected into the dlPAG or amv. Mice were implanted unilaterally with fiberoptic cannulae in the PMd, amv, dlPAG. A 400 µm diameter, 0.48 NA optical fiber (Neurophotometrics) was used for photometry experiments. Adhesive cement (C&B metabond; Parkell, Edgewood, NY) and dental cement (Stoelting, Wood Dale, IL) were used to securely attach the fiber optic cannula to the skull. For miniaturized microscope experiments, 40 nl of AAV9-DIO-GCaMP6s was injected in the PMd of cck-cre mice and a 7 mm GRIN lens was implanted 200 µM above the infusion site. Three weeks following surgery, animals were base-plated. For dual photometry recordings, injections and fiber implants were done in the same mouse contralaterally to record PMd-cck and dlPAG-syn cell bodies simultaneously.

## Rat exposure assay

Mice were accustomed to handling prior to any behavioral assay. On day 1, mice were habituated to a rectangular box (70 cm length, 26 cm width, 44 cm height) for 20 min. This environment consisted of a large aquarium made of glass. Sheets of paper lined the outside glass surface. The box was cleaned with ethanol between mice. Twenty-four hours later, mice were exposed to the same environment but in the presence of a toy rat for 20 min. Mice were then exposed to an adult rat or a toy rat in this environment on the two following days. The rat was secured by a harness tied to one of the walls and could freely ambulate only within a short radius of approximately 20 cm. The mouse was placed near the wall opposite to the rat and freely explored the context for 20 min. No separating barrier was placed between the mouse and the rat, allowing for close naturalistic encounters that can induce a variety of robust defensive behaviors.

## Contextual fear conditioning test

To better evaluate a broader species-specific defense repertoire in face of a conditioned stimulus, we used a modified version of the standard contextual fear conditioning method (*Schuette et al., 2020*). Pre-shock, fear conditioning, and retrieval sessions were performed in a context (70 cm length × 17 cm width × 40 cm height) with an evenly distributed light intensity of 40 lux and a Coulbourn shock grid (19.5 cm × 17 cm) set at the extreme end of the enclosure. The fear conditioning environment is made of laminated white foam board. The box was cleaned with ethanol between mice. Forty-eight hours after rat exposure, mice were habituated to this context and could freely explore the whole

environment for 20 min. On the following day, the grid was activated, such that a single 0.7 mA foot shock was delivered for 2 s only on the first time the mouse fully entered the grid zone. Twenty-four hours later, retrieval sessions were performed in the same enclosure but without shock. Mice could freely explore the context for 20 min during pre-shock habituation, fear conditioning, and retrieval sessions.

## Behavioral quantification

To extract the pose of freely behaving mice in the described assays, we implemented DeepLabCut (*Nath et al., 2019*), an open-source convolutional neural network-based toolbox, to identify mouse nose, ear, and tailbase xy-coordinates in each recorded video frame. These coordinates were then used to calculate velocity and position at each timepoint, as well as classify behaviors such as escape runs and freezes in an automated manner using custom Matlab scripts. Specifically:

'Escapes' were defined as epochs for which (1) the mouse speed away from the threat or control threat exceeded 2 cm/s (as there was little room for acceleration between the threat zone and opposite wall, the speed threshold was set to this relatively low value), (2) movement away from the threat was initiated at a minimum distance-from-threat of 30 cm, and (3) the distance traversed from escape onset to offset was greater than 10 cm. Thus, escapes were required to begin near the threat and lead to a substantial increase in distance from the threat.

'Escape speed' was defined as the average speed from escape onset to offset.

'Escape angle' was defined as the cosine of the mouse head direction in radians, such that the values ranged from –1 (facing toward the threat) to 1 (facing away from the threat). Mouse head direction was determined by the angle of the line connecting a point midway between the ears and the nose.

'Approaches' were defined as epochs for which (1) the mouse speed toward the threat or control threat exceeded 2 cm/s and (2) the distance traversed from approach onset to offset was greater than 10 cm.

'Stretch-attend postures' were defined as epochs for which (1) the distance between mouse nose and tailbase exceeded a distance of approximately 1.2 mouse body lengths and (2) mouse tailbase speed fell below 1 cm/s.

'Freezes' were defined as periods for which mouse nose and tailbase speed fell below 0.25 cm/s for at least 0.33 s (*Schuette et al., 2020*). 'Freeze bout duration' was defined as the amount of time that elapsed from freeze onset to offset.

'Walks' were defined as epochs for which (1) movements along the safe wall of the enclosure, perpendicular to the threat, exceeded 2 cm/s and (2) the distance traversed from walk onset to offset was greater than 5 cm.

All behaviors were manually checked by the experimenters for error.

## Behavioral protocols

The order of assays was identical for behavioral characterization (*Figure 1*), fiber photometry (*Figures 2, 7 and 9*) and miniscope (*Figure 3*) experiments, as detailed in *Figures 1 and 2C*. Specifically, mice were habituated to the rat enclosure for days 1–3. The toy rat and live rat were introduced on days 4 and 5. This was followed by habituation to the fear conditioning enclosure on days 6–8. Day nine was the pre-shock control session. Fear acquisition and retrieval were performed, respectively, on days 10 and 11.

The DREADD experiments were performed as diagrammed in *Figure 5C*. Saline or CNO were administered on contiguous days. Habituation to the rat enclosure occurred on days 1–3. The toy rat was introduced on days 4 (saline) and 5 (CNO) and the live rat on days 8 (saline) and 9 (CNO). This was followed by habituation to the fear conditioning enclosure on days 11–13. Days 14 (saline) and 15 (CNO) were considered the pre-shock sessions. Fear acquisition occurred on day 16, followed by fear retrieval on days 17 (saline) and 18 (CNO).

## Fiber photometry

Photometry was performed as described in detail previously (*Kim et al., 2016*). Briefly, we used a 405 nm LED and a 470 nm LED (Thorlabs, M405F1 and M470F1) for the $Ca^{2+}$-dependent and $Ca^{2+}$-independent isosbestic control measurements. The two LEDs were bandpass filtered (Thorlabs, FB410-10

and FB470-10) and then combined with a 425 nm longpass dichroic mirror (Thorlabs, DMLP425R) and coupled into the microscope using a 495 nm longpass dichroic mirror (Semrock, FF495-Di02−25 × 36). Mice were connected with a branched patch cord (400 µm, Doric Lenses, Quebec, Canada) using a zirconia sleeve to the optical system. The signal was captured at 20 Hz (alternating 405 nm LED and 470 nm LED). To correct for signal artifacts of a nonbiological origin (i.e., photobleaching and movement artifacts), custom Matlab scripts leveraged the reference signal (405 nm), unaffected by calcium saturation, to isolate and remove these effects from the calcium signal (470 nm).

## Fiber photometry behavior-triggered averaging
To plot the behavior-triggered averages, only mice that displayed a minimum of three behavioral instances were included in the corresponding behavioral figure. Moreover, event-triggered averages were only calculated from behavioral instances that were separated from other classified behavioral instances by a minimum of 5 s.

## Miniscope video capture
All videos were recorded at 30 frames/s using a Logitech HD C310 webcam and custom-built head-mounted UCLA miniscope (*Cai et al., 2016*). Open-source UCLA Miniscope software and hardware (http://miniscope.org/) were used to capture and synchronize neural and behavioral video (*Cai et al., 2016*).

## Miniscope postprocessing
The open-source UCLA miniscope analysis package (https://github.com/daharoni/Miniscope_Analysis, *Daniel, 2021*, *Aharoni and Hoogland, 2019*) was used to motion correct miniscope videos. They were then temporally downsampled by a factor of four and spatially downsampled by a factor of two. The cell activity and footprints were extracted using the open-source package Constrained Nonnegative Matrix Factorization for microEndoscopic data (CNMF-E; https://github.com/zhoupc/CNMF_E, *Pengcheng, 2021*, *Schuette et al., 2020*; *Zhou et al., 2018*). Only cells whose variance was greater than or equal to 25 % of the maximum variance among non-outliers were used in the analysis.

## Behavior decoding using PMd neural data
Discrete classification of escape behavior was performed using multinomial logistic regression. Timepoints following escape by 2 s were labeled 'escape', and a matched number of non-escape timepoints were randomly selected for training and validation. Each time point was treated as an individual data point. Training and validation were performed using fivefold cross-validation, with a minimum of 10 s between training and validation sets. As equal numbers of escape and non-escape samples were used to build the training and validation sets, chance accuracy was 50 %. Sessions with less than five escapes were excluded from the analysis. The same analysis was performed for approach, stretch-attend postures, and freeze. To predict escape at negative time lags from behavior onset, the same analysis procedure was implemented, using 2 s epochs preceding escape by 2, 4, 6, 8, and 10 s.

## Behavior cell classification
We used a GLM to identify cells that showed increased calcium activity during approach, stretch-attend, escape, and freeze behaviors. We fit this model to each cell's activity, with behavior indices as the predictor variable and behavior coefficients as the measure of fit. Behavior onset times were then randomized 100 times and a bootstrap distribution built from the resulting GLM coefficients. A cell was considered a behavior-categorized cell if its coefficient exceeded 95 % of the bootstrap coefficient values.

## Escape speed cell classification
Escape speed cells were classified using the method described in *Iwase et al., 2020*. Briefly, we calculated the Pearson product-moment correlation coefficient between each cell's firing rate and the animal's running speed during escape. The chance distribution was determined using a shuffling procedure whereby the calcium data was time-shifted in a circular manner relative to speed by a random duration between 30 s and the total duration of the assay minus 30 s. This was repeated 100 times for each cell. Thus, a cell was categorized as a 'escape speed-correlated cell' if the absolute

value of its Pearson product-moment correlation exceeded the 95th percentile of distribution of speed scores from the chance distribution of all cells recorded in the PMd.

### Position and speed decoding

To predict position and speed from neural data, the data dimensionality was reduced by principal component analysis, such that the top principal components, representing at least 80 % of the total variance, were used in the following decoding analysis. This output and the related position/speed data were then separated into alternating 60 s training and testing blocks, with 10 s of separation between blocks. Odd blocks were used to train a generalized linear regression model (GLM; Matlab function 'glmfit') and withheld even blocks were used to test the resulting model. Accuracies of this withheld testing block were reported as mean squared error. The level of chance error was calculated as the mean testing error of the GLM on circularly permuted data (100 iterations per session) across animals.

### Mutual information analyses

Mutual information is an information theory-derived metric reflecting the amount of information obtained for one variable by observing another variable. In the case of the fiber photometry analysis, the related variables were the simultaneously recorded PMd and dlPAG signals. Mutual information was calculated using custom Matlab code (*Delpiano, 2021*) for all samples where the speed was greater than 1 cm/s. Calculating mutual information requires computing the joint distribution over the PMd and dlPAG fiber photometry signals. This distribution was calculated using a histogram count after discretizing PMd and dlPAG fiber photometry signals each into 20 bins. The same approach was used for the miniscope mutual information analysis, for which this metric was computed for all escape samples between the calcium signal of individual PMd-cck cells and speed.

### Chemogenetics

Mice used for chemogenetic experiments were exposed to each threat and control stimuli twice, once following treatment with saline and once following treatment with CNO (5 mg/kg, injected intraperitoneally) 40 min prior to the experiment. Only one control or threat-exposure assay was performed per day with each mouse.

### Behavior video capture

All behavior videos were captured at 30 frames/s in standard definition (640 × 480) using a Logitech HD C310 webcam. To capture fiber-photometry synchronized videos, both the calcium signal and behavior were recorded by the same computer using custom Matlab scripts that also collected time-stamp values for each calcium sample/behavioral frame. These timestamps were used to precisely align neural activity and behavior.

### Light delivery for optogenetics

For PMd-cck ChR2 mice, blue light was generated by a 473 nm laser (Dragon Lasers, Changchun Jilin, China) at 4.5 mW unless otherwise indicated. Green light was generated by a 532 nm laser (Dragon Lasers), and bilaterally delivered to mice at 10 mW. A Master-8 pulse generator (A.M.P.I., Jerusalem, Israel) was used to drive the blue laser at 20 Hz. This stimulation pattern was used for all ChR2 experiments. The laser output was delivered to the animal via an optical fiber (200 µm core, 0.22 numerical aperture, Doric Lenses, Canada) coupled with the fiberoptic implanted on the animals through a zirconia sleeve.

### Immunostaining for Cfos

Fixed brains were kept in 30 % sucrose at 4 °C overnight and then sectioned on a cryostat (40 µm) slices. Sections were washed in PBS and incubated in a blocking solution (3 % normal donkey serum and 0.3 % triton-x in PBS) for 1 hr at room temperature. Sections were then incubated at 4 °C for 12 hr with polyclonal anti-fos antibody made in rabbit (1/500 dilution) (c-Fos (9F6) Rabbit mAb CAT#2250, Cell Signalling Technology) in blocking solution. Following primary antibody incubation, sections were washed in PBS three times for 10 min and then incubated with anti-rabbit IgG (H + L) antibody (1/500 dilution) conjugated to Alexa Fluor 594 (red) (CAT# 8,889 S, cellsignal.com) for 1 hr at room

temperature. Sections were washed in PBS three times for 10 min, incubated with DAPI (1/50,000 dilution in PBS), washed again in PBS, and mounted in glass slides using PVA-DABCO (Sigma).

## Perfusion and histological verification

Mice were anesthetized with Fatal-Plus and transcardially perfused with phosphate buffered saline followed by a solution of 4 % paraformaldehyde. Extracted brains were stored for 12 hr at 4 °C in 4 % paraformaldehyde. Brains were then placed in sucrose for a minimum of 24 hr. Brains were sectioned in the coronal plane in a cryostat, washed in phosphate buffered saline, and mounted on glass slides using PVA-DABCO. Images were acquired using a Keyence BZ-X fluorescence microscope with a 10 or 20 × air objective.

## Acute brain slice preparation and electrophysiological recordings

Cck-cre driver line mice were injected with AAV9-FLEX-ChR2-YFP in the PMd. Acute slices were prepared from these mice. For electrophysiological measurements, slices were transferred as needed to the recording chamber, where they were perfused with oxygenated aCSF at 32 °C. The slices were held in place using a nylon net stretched within a U-shaped platinum wire. Visually guided whole-cell patch clamp recordings were made using infrared differential interference contrast optics. We also verified the identity of PMD neurons by only recording from YFP-positive neurons. All recordings were obtained using a MultiClamp 700B amplifier system (Molecular Devices, Union City, CA). Experiments were controlled by PClamp 10 software running on a PC, and the data were acquired using the Digidata 1,440A acquisition system. All recording electrodes (3–8 MΩ) were pulled from thin-walled capillary glass (A-M Systems, Carlsborg, WA) using a Sutter Instruments P97 puller. The patch pipettes were filled with internal solution containing (in mM) 100 K- gluconate, 20 KCl, 4 ATP-Mg, 10 phospho-creatine, 0.3 GTP-Na, and 10 HEPES (in mM) with a pH of 7.3 and osmolarity of 300 mOsm. Only cells with a stable, uncorrected resting membrane potential (RMP) between –50 and –80 mV, overshooting action potentials, and an input resistance (RN) >100 MW were used. To minimize the influence of voltage-dependent changes on membrane conductances, all cells were studied at a membrane potential near –60 mV (using constant current injection under current clamp mode). To study intrinsic firing properties of PMD neurons, WCRs were conducted under current clamp using the following protocol: (1) Voltage–current (V-I) relationships were obtained using 400 ms current steps (range –50 pA to rheobase) and by plotting the plateau voltage deflection against current amplitude. Neuronal input resistance (RN) was determined from the slope of the linear fit of that portion of the V-I plot where the voltage sweeps did not exhibit sags or active conductance. (2) Intrinsic excitability measurements were obtained using 1 s current steps (range 0–500 pA) and by plotting the number of action potentials fired against current amplitude. (3) RMP was calculated as the difference between mean membrane potential during the first minute immediately after obtaining whole cell configuration and after withdrawing the electrode from the neuron.

For validating hM4Di in PMd-cck cells, acute brain slices preparation and electrophysiological recordings were performed using standard methods as previously described (*Nagai et al., 2019*). Briefly, Cck-Cre+ mice that had received AAV microinjections into PMd were deeply anesthetized with isoflurane and decapitated with sharp shears. The brains were placed and sliced in ice-cold modified artificial CSF (aCSF) containing the following (in mM): 194 sucrose, 30 NaCl, 4.5 KCl, 1 MgCl$_2$, 26 NaHCO$_3$, 1.2 NaH2PO$_4$, and 10 D-glucose, saturated with 95 % O$_2$ and 5 % CO$_2$. A vibratome (DSK-Zero1) was used to cut 300 μm brain sections. The slices were allowed to equilibrate for 30 min at 32°C–34°C in normal aCSF containing (in mM); 124 NaCl, 4.5 KCl, 2 CaCl$_2$, 1 MgCl$_2$, 26 NaHCO$_3$, 1.2 NaH$_2$PO$_4$, and 10 D-glucose continuously bubbled with 95 % O$_2$ and 5 % CO$_2$. Slices were then stored at 21°C–23°C in the same buffer until use. All slices were used within 2–6 hr of slicing.

Slices were placed in the recording chamber and continuously perfused with 95 % O$_2$ and 5 % CO$_2$ bubbled normal aCSF. pCLAMP10.4 software and a Multi-Clamp 700B amplifier was used for electrophysiology (Molecular Devices). Whole-cell patch-clamp recordings were made from neurons in the PMd or dorsolateral PAG (dlPAG) using patch pipettes with a typical resistance of 4–5 MΩ. Neurons were selected based on reporter fluorescence (mCherry for hM4Di-mCherry). The intracellular solution for recordings comprised the following (in mM): 135 potassium gluconate, 5 KCl, 0.5 CaCl$_2$, 5 HEPES, 5 EGTA, 2 Mg-ATP, and 0.3 Na-GTP, pH 7.3 adjusted with KOH. The initial access resistance values were <20 MΩ for all cells; if this changed by >20 %, the cell was discarded. Light flashes (0.2 mW/

mm²) from a blue LED light source (Sutter Instruments) were delivered via the microscope optics and a 40 × water immersion objective lens and controlled remotely using TTL pulses from Clampex. Cell responses were recorded in whole-cell mode and recorded using an Axopatch 700B amplifier connected via a digitizer to a computer with pCLAMP10 software. To stimulate ChR2 expressed in PMd neurons or axons, 5 ms pulses were delivered at inter-pulse intervals of 200 ms, 50 ms, or 25 ms for 5, 20, or 40 Hz optical stimulations, respectively.

## Functional magnetic resonance imaging methods

### Participants
This study included 48 adult participants (mean ± SD age: 25.1 ± 7.1; 27 male, 21 female; seven left-handed; 40 white and eight non-white [one Hispanic, five Asian, one Black, and one American Indian]). All participants were healthy, with normal or corrected to normal vision and normal hearing, and with no history of psychiatric, physiological, or pain disorders and neurological conditions; no current pain symptoms; and no MRI contraindications. Eligibility was assessed with a general health questionnaire, a pain safety screening form, and an MRI safety screening form. Participants were recruited from the Boulder/Denver Metro Area. The institutional review board of the University of Colorado Boulder approved the study, and all participants provided written informed consent.

### Experimental paradigm
Participants received five different types of aversive stimulation (mechanical pain, thermal pain, aversive auditory, aversive visual, and pleasant visual), each at four stimulus intensities. Twenty-four stimuli of each type (six per intensity) were presented over six fMRI runs in random order. Following stimulation on each trial, participants made behavioral ratings of their subjective experience. Participants were instructed to answer the question 'How much do you want to avoid this experience in the future?'. Ratings were made with a non-linear visual analog rating scale, with anchors 'Not at all' and 'Most' displayed at the ends of the scale.

### Stimuli
Visual stimulation was administered on the MRI screen and included normed images from the International Affective Picture System (IAPS) database (*Lang et al., 2008*). To induce four 'stimulus intensity levels' we selected four groups of seven images based on their normed aversiveness ratings (averaged across male and female raters) available in the IAPS database and confirmed by N = 10 lab members (five male, five female) in response to 'How aversive is this image? 1–100'. Selected images included photographs of animals (n = 7), bodily illness and injury (n = 12), and industrial and human waste (n = 9). Four stimulus levels were delivered to participants for 10 s each.

### MRI data acquisition and preprocessing
Whole-brain fMRI data were acquired on a 3T Siemens MAGNETOM Prisma Fit MRI scanner at the Intermountain Neuroimaging Consortium facility at the University of Colorado, Boulder. Structural images were acquired using high-resolution T1 spoiled gradient recall images (SPGR) for anatomical localization and warping to standard MNI space. Functional images were acquired with a multiband EPI sequence (TR = 460 ms, TE = 27.2 ms, field of view = 220 mm, multiband acceleration factor = 8, flip angle = 44°, 64 × 64 image matrix, 2.7 mm isotropic voxels, 56 interleaved slices, phase encoding posterior >> anterior). Six runs of 7.17 min duration (934 total measurements) were acquired. Stimulus presentation and behavioral data acquisition were controlled using Psychtoolbox.

fMRI data were preprocessed using an automated pipeline implemented by the Mind Research Network, Albuquerque, NM. Briefly, the preprocessing steps included distortion correction using FSL's top-up tool (https://fsl.fmrib.ox.ac.uk/fsl/), motion correction (affine alignment of first EPI volume [reference image]) to T1, followed by affine alignment of all EPI volumes to the reference image and estimation of the motion parameter file (sepi_vr_motion.1D, AFNI, https://afni.nimh.nih.gov/), spatial normalization via subject's T1 image (T1 normalization to MNI space [nonlinear transform]), normalization of EPI image to MNI space (3dNWarpApply, AFNI, https://afni.nimh.nih.gov/), interpolation to 2 mm isotropic voxels, and smoothing with a 6 mm FWHM kernel (SPM 8, https://www.fil.ion.ucl.ac.uk/spm/software/spm8/).

Prior to first-level (within-subject) analysis, we removed the first four volumes to allow for image intensity stabilization. We also identified image-wise outliers by computing both the mean and the standard deviation (across voxels) of intensity values for each image for all slices to remove intermittent gradient and severe motion-related artifacts (spikes) that are present to some degree in all fMRI data.

## fMRI data analysis

Data were analyzed using SPM12 (http://www.fil.ion.ucl.ac.uk/spm) and custom MATLAB (The MathWorks, Inc, Natick, MA) code available from the authors' website (http://github.com/canlab/CanlabCore, *Wager, 2021*). First-level GLM analyses were conducted in SPM12. The six runs were concatenated for each subject. Boxcar regressors, convolved with the canonical hemodynamic response function, were constructed to model periods for the 10 s stimulation and 4–7 s rating periods. The fixation cross epoch was used as an implicit baseline. A high-pass filter of 0.008 Hz was applied. Nuisance variables included (1) 'dummy' regressors coding for each run (intercept for each run); (2) linear drift across time within each run; (3) the six estimated head movement parameters (x, y, z, roll, pitch, and yaw), their mean-centered squares, their derivatives, and squared derivative for each run (total 24 columns); and (4) motion outliers (spikes) identified in the previous step. A 'single-trial model' was used to uniquely estimate the response to every stimulus in order to assess functional connectivity.

## Functional connectivity analysis

Functional connectivity between the hypothalamus and PAG was estimated using PLS (*Wold et al., 2001*) regression, which identifies latent multivariate patterns that maximize the covariance between two blocks of data (i.e., BOLD activity in hypothalamus and PAG voxels). Here, data comprised single trial estimates of brain activation in response to aversive thermal, mechanical, auditory, and visual stimuli, in addition to a set of pleasant visual stimuli that were used as a control. For the PLS model, the predictor block of variables included all voxels in an anatomically defined mask of the hypothalamus (*Pauli et al., 2018*) (337 voxels) and the outcome block included all voxels in the PAG (*Kragel et al., 2019*) (42 voxels). Localization of the hypothalamus signal that covaries with the PAG responses was performed by bootstrapping the PLS regression and examining the distribution of PLS regression coefficients and their deviation from zero (using normal approximation for inference). Hyperalignment of fMRI data (*Haxby et al., 2011*) was conducted separately for each region as a preprocessing step, and leave-one-subject-out cross-validation was performed to estimate the strength of functional connections (i.e., the Pearson correlation between the first 'X score' and 'Y score' estimated by PLS), similar to the canonical correlation (*Hardoon et al., 2004*).

A benefit of the pathway-identification model we employed is that it can, in principle, identify HTH and PAG patterns that distinctly participate in the HTH–PAG pathway. For example, the central nucleus of the amygdala (CeA) projects to both the hypothalamus and the PAG (*Kim et al., 2013*) and could indirectly explain variation in BOLD signals in the PAG. To test pathway specificity, we separately modeled a pathway between the CeA and the PAG using the approach described above. This allowed us to evaluate how much variation in PAG activity the HTH–PAG pathway explained above and beyond the CeA–PAG pathway. To evaluate this, we computed the partial correlation between latent sources in the hypothalamus and PAG, controlling for the latent source in the CeA.

## Statistics

Nonparametric Wilcoxon signed-rank or rank-sum tests were used, unless otherwise stated. Two-tailed tests were used throughout with $\alpha$ = 0.05. Non-parametric tests were used because normality tests are severely underpowered for n < 100, indicating that, with small n, normality tests will often fail to detect non-normal distributions (*Razali and Wah, 2011*). However, by necessity, rodent cohorts are much smaller than n = 100. Thus, to avoid unwarranted normality assumptions about our data, we used non-parametric tests. Asterisks in the figures indicate the p values. Standard error of the mean was plotted in each figure as an estimate of variation. Multiple comparisons were adjusted with the false discovery rate (FDR) method.

## Behavioral cohort information

Initial behavioral characterization of the assays (*Figure 1*) was replicated three times, with cohorts containing 10, 10, and 12 mice (32 in total). PMd cell body fiber photometry experiments (*Figure 2*) were replicated twice with cohorts containing 7 and 8 mice (15 in total). Miniscope experiments (*Figures 3 and 4*) were replicated twice, with cohorts of four and five mice (nine in total). Chemogenetics experiments (*Figure 5*) were replicated twice (cohort 1 with 10 controls and 6 hM4Di mice and cohort 2 with 9 controls and 5 hM4Di mice). ChR2 experiments (*Figure 6*) were done once, with five YFP and four ChR2 mice. dlPAG fiber photometry experiments (*Figure 7*) were replicated twice, with cohorts of four and five mice (nine in total). Amv body fiber photometry experiments (*Figure 7*) were replicated once with six mice. PMd–dlPAG optogenetic projection inhibition experiments (*Figure 8*) were replicated twice. Both cohorts had 12 controls and six arch mice. PMd–amv optogenetic projection inhibition experiments (*Figure 8*) were replicated twice. Both cohorts had six controls and nine arch mice. Appropriate fluorophore-only expressing mice were used as controls for chemogenetic and optogenetic experiments. For fMRI data (*Figure 9*), a cohort of 48 human subjects was used only once. Each mouse was only exposed to each assay once, as defensive behavior assays cannot be repeated. Thus, there are no technical replicates. No outliers were found or excluded. All mice and humans were used. Sample sizes for human and mouse experiments were determined based on comparisons to similar published papers.

For chemogenetic and optogenetic experiments, mice in each cage were randomly allocated to control (mcherry or YFP -expressing mice) or experimental conditions (hM4Di-, ChR2-, or Arch-expressing mice). Data collection was done blinded to treatment group in mice. For human fMRI data and mouse neural activity recordings, all data were obtained from subjects in identical conditions, and thus they were all allocated to the same experimental group. There were no experimentally controlled differences across these subjects and, thus, there were no 'treatment groups'.

## Data and code availability

Custom analysis scripts are available at https://github.com/schuettepeter/PMd_escape_vigor (copy archived at swh:1:rev:5a9232a5dee602fa57cb1e959f63c10da91cd1db, *Schuette, 2021*). Data is available at https://datadryad.org/stash/share/dYuSl2nnXsyi0nTDjCDeHR08gwW7paFL4Eo3TmF_aH4.

## Acknowledgements

We were supported by the NIMH (R00 MH106649 and R01 MH119089) (AA), the Brain and Behavior Research Foundation (Grants # 22663, 27654, 27780, and 29204, respectively, to AA, FMCVR, WW, and JCK), the NSF (NSF-GRFP DGE-1650604, PJS), the UCLA Affiliates fellowship (PJS), the Achievements Rewards for College Scientists Foundation, NIMH (F31 MH121050-01A1) (MQL), the Hellman Foundation (AA), and FAPESP (Research Grant #2014/05432–9) (NSC). FMCVR was supported with FAPESP grants #2015/23092–3 and #2017/08668–1. We thank Profs. BSK and JN for performing patch-clamping experiments.

## Additional information

### Funding

| Funder | Grant reference number | Author |
| --- | --- | --- |
| National Institutes of Health | R00 MH106649 | Avishek Adhikari |
| National Institutes of Health | R01 MH119089 | Avishek Adhikari |
| Brain and Behavior Research Foundation | 22663 | Avishek Adhikari |
| Brain and Behavior Research Foundation | 27654 | Fernando MCV Reis |

| Funder | Grant reference number | Author |
|---|---|---|
| Brain and Behavior Research Foundation | 27780 | Weisheng Wang |
| Brain and Behavior Research Foundation | 29204 | Jonathan C Kao |
| National Institutes of Health | F31 MH121050-01A1 | Mimi Q La-Vu |
| National Science Foundation | DGE-1650604 | Peter J Schuette |
| Fundação de Amparo à Pesquisa do Estado de São Paulo | 2015/23092-3 | Fernando MCV Reis |
| Fundação de Amparo à Pesquisa do Estado de São Paulo | 2017/08668-1 | Fernando MCV Reis |
| Fundação de Amparo à Pesquisa do Estado de São Paulo | 2014/05432-9 | Newton S Canteras |
| Hellman Foundation | | Avishek Adhikari |
| Achievement Rewards for College Scientists Foundation | | Mimi Q La-Vu |

The funders had no role in study design, data collection and interpretation, or the decision to submit the work for publication.

## Author contributions

Weisheng Wang, Conceptualization, Data curation, Funding acquisition, Investigation, Methodology, Writing - original draft, Writing - review and editing; Peter J Schuette, Conceptualization, Formal analysis, Software, Writing - original draft, Writing - review and editing; Mimi Q La-Vu, Funding acquisition, Investigation; Anita Torossian, Investigation, Methodology; Brooke C Tobias, Investigation; Marta Ceko, Data curation, Formal analysis, Investigation, Methodology, Software, Writing - review and editing; Philip A Kragel, Formal analysis, Investigation, Methodology, Software, Writing - review and editing; Fernando MCV Reis, Conceptualization, Funding acquisition, Investigation, Methodology; Shiyu Ji, Data curation; Megha Sehgal, Meghmik Chakerian, Data curation, Methodology; Sandra Maesta-Pereira, Data curation, Methodology, Software; Alcino J Silva, Conceptualization, Funding acquisition; Newton S Canteras, Conceptualization, Funding acquisition, Writing - review and editing; Tor Wager, Conceptualization, Formal analysis, Funding acquisition, Investigation, Methodology, Supervision; Jonathan C Kao, Formal analysis, Funding acquisition, Methodology, Software; Avishek Adhikari, Conceptualization, Funding acquisition, Writing - original draft, Writing - review and editing

## Author ORCIDs

Peter J Schuette (iD) http://orcid.org/0000-0002-6308-6441
Brooke C Tobias (iD) http://orcid.org/0000-0003-2043-9523
Fernando MCV Reis (iD) http://orcid.org/0000-0002-0121-2887
Shiyu Ji (iD) http://orcid.org/0000-0003-3413-5766
Sandra Maesta-Pereira (iD) http://orcid.org/0000-0001-6522-8311
Alcino J Silva (iD) http://orcid.org/0000-0002-1587-4558
Newton S Canteras (iD) http://orcid.org/0000-0002-7205-5372
Jonathan C Kao (iD) http://orcid.org/0000-0002-9298-0143
Avishek Adhikari (iD) http://orcid.org/0000-0002-9187-9211

## Ethics

All procedures have been approved by the University of California, Los Angeles Institutional Animal Care and Use Committee, protocols 2017-011 and 2017-075.

## Decision letter and Author response

Decision letter https://doi.org/10.7554/eLife.69178.sa1

Author response https://doi.org/10.7554/eLife.69178.sa2

## Additional files

### Supplementary files
• Transparent reporting form

### Data availability
All custom written software has been uploaded to https://github.com/schuettepeter/PMd_escape_vigor (copy archived at swh:1:rev:5a9232a5dee602fa57cb1e959f63c10da91cd1db) Data has been uploaded to https://doi.org/10.5068/D19H5X.

The following dataset was generated:

| Author(s) | Year | Dataset title | Dataset URL | Database and Identifier |
|---|---|---|---|---|
| Schuette P | 2021 | Data from: Dorsal premammillary projection to periaqueductal gray controls escape vigor from innate and conditioned threats | https://doi.org/10.5061/D19H5X | Dryad Digital Repository, 10.5061/dryad.D19H5X |

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
