## [Decision Letter]

**Acceptance summary:**

This manuscript demonstrates the function of a hypothalamic-dorsal PAG pathway in escape behavior. The experiments presented here add to a growing body of evidence by exploring the role of this projection in behavioral retreat from associative and non-associative aversive stimuli, using approaches that quantify neural activity alongside causal manipulations. Novel features of this report include its emphasis on escape vigor, the establishment of predictive relationships between neural activity and escape, and the interdisciplinary nature of the data presented. Additionally, the response to initial reviews was thoughtful and thorough. Well done and congrats on an intriguing paper!

**Decision letter after peer review:**

Thank you for submitting your article "Dorsal premammillary projection to periaqueductal gray controls escape vigor from innate and conditioned threats" for consideration by *eLife*. Your article has been reviewed by 3 peer reviewers, one of whom is a member of our Board of Reviewing Editors, and the evaluation has been overseen by Laura Colgin as the Senior Editor. The following individual involved in review of your submission has agreed to reveal their identity: Jonathan Fadok (Reviewer #3).

Essential revisions:

1) The methods and results need to be presented with greater clarity and specificity. The reviewers had substantial questions about how the manuscript dealt with behavioral definitions and metrics, experimental design, and the conclusions drawn about the relationship between behavioral outcomes and neuronal data. In some cases a thorough rewrite is needed, but in others new analyses may be required (see below).

2) The results presented here must be discussed in the context of other recent findings from the same lab, such as a Neuron paper published by the primary authors of this report (for instance, the Neuron work could provide a rationale for the targeting cck cells that is otherwise missing, and also sheds some light on the negative results obtained for dPM-amv projections).

3) The authors should consider dropping the fMRI results. Though potentially interesting in and of themselves, the relationship between passive viewing of aversive images and escape behavior is not clear. The reviewers felt that these data reduce clarity rather than add insight into what, at best, is a hypothetical homology.

*Reviewer #1 (Recommendations for the authors):*

A recent publication in Neuron with the same 1st and last authors (Wang et al., 2021, Neuron, 109) seems to substantially scoop this manuscript. Most of the basic finds presented here are conceptual replications of what has been recently demonstrated in Neuron, where the primary authors of this submission to *eLife* report that dorsal PAG-projecting cck neurons in PMd are activated by and required for escape behavior. The Neuron publication significantly curtails the novelty of this work. The current manuscript does make a few incremental advances. It delves into the predictive relationship between Ca++ transients in PMd cck neurons and escape to a greater degree than the Neuron paper (though the Neuron paper indeed demonstrates such a relationship). It also shows that escape vigor (ie velocity) is mediated by PMd cck circuits (though the fundamental role of this circuitry in escape is established in the Neuron paper). Because of their fundamental similarity, these results submitted to *eLife* must be discussed in terms of the results published in Neuron so that it is clear to the reader exactly how the current work advances the understanding of escape circuits.

A paper cited by the authors (Cezario et al., 2008) shows a causal role for PMd in defensive reactions to a context previously paired with a predator (in which the predator essentially acts as a US). While this report did not examine escape per se, it does further limit the novelty of what is demonstrated here. It needs to be made clear that a specific role of PMd in quantitative measures of escape is the new result, instead of a broader role role for this region and related circuits in the defensive response to innate and conditioned threats (which has already been established, including by the authors).

It is unclear why the behavioral assays were not counterbalanced. It is possible that recent exposure to predator alters the subjects' response to the aversive stimuli and contextual cues via sensitization or some other process. This weakens the conclusions that can be drawn about the role of premammillary-dPAG projections in behavioral withdrawal from conditioned stimuli associated with shock, thus decreasing the impact of these results.

The meaning of the fMRI data is unclear. The authors claim that it is not possible to examine the escape behavior in the scanner, but there is a history of avoidance/escape paradigms using fMRI (eg Mobbs et al., 2007, Science, 317). For instance, a simple button press to remove aversive images could serve as an escape response. Absent any such behavioral measure, it is not really possible to understand how hypothalamic-PAG BOLD responses to passive viewing of aversive images relate to the circuit level data obtained in mice. If anything, activity in these regions in the absence of escape behavior seems to belie the central point of the paper.

*Reviewer #2 (Recommendations for the authors):*

While we enjoyed the manuscript overall, we often encountered difficulties in understanding exactly what was done experimentally. We therefore strongly suggest overhauling the methods section to better and unambiguously describe all methodology, with a focus on behavioral analyses. In addition, we would suggest a more careful and comprehensive interpretation of the characterized pathway, leaving room for potential broader functionality. This in our view would take into account the fact that this pathways presents one circuit element within a much larger network, which certainly subserves different functions in a rather dynamic manner.

*Reviewer #3 (Recommendations for the authors):*

1. Please provide more specific information in the methods about how the rat exposure and fear conditioning contexts are differentiated. For example, are they made from the same material? Are they cleaned with different solutions?

2. Non-parametric tests are used throughout to determine significance. The rationale behind this decision should be explicitly stated. Were data tested for normality?

3. Terminology for control conditions is inconsistent throughout the figures, e.g. 'control/pre-shock/habituation.'

4. The electrophysiological findings are not problematic, but do not strengthen the paper. Without knowing the relative strength of excitatory inputs to the CCK neurons, the excitability measures are not interpretable.

5. The titer of the inhibitory DREADD virus is not reported.

6. What is the rationale for the prolonged optogenetic excitation (10 min at 20Hz)?

---

## [Author Response]

Essential revisions:1) The methods and results need to be presented with greater clarity and specificity. The reviewers had substantial questions about how the manuscript dealt with behavioral definitions and metrics, experimental design, and the conclusions drawn about the relationship between behavioral outcomes and neuronal data. In some cases a thorough rewrite is needed, but in others new analyses may be required (see below).

We have rewritten the Methods section with detailed information on the behavioral definitions and metrics used (see Methods section “Behavioral Quantification”). We also clarified the experimental design and performed other changes as requested by the Reviewers.

We have added detailed information on the definition of each of the scored defensive behaviors. We also performed significant additional analysis. Detailed descriptions of these new analyses and experiments are written in point-by-point responses to the Reviewers below. Briefly, we showed the following:

– Prior experience with the Rat assay does not change any behavioral metric in the footshock assay (Figure R1)(same as figure 1, figure supplement 2).

– Mutual information between PMd-cck and dlPAG-syn cells increase during exposure to threats compared to control (Figure R2). (same as Figure 9).

– PMd-cck neural activity is higher during exposure to threat compared to control over regardless of the speed range analyzed (Figure R3) (same as Figure 2, figure supplement 2).

– The angle of escape trajectory is highly conserved (Figure R4).

– We showed detailed characterization of escape-related metrics for animals used for fiber photometry recordings (Figure R5) (same as figure 2, figure supplement 1).

– There is no overlap between scored behaviors (Figure R6).

– We plotted individual mouse data for the defensive behaviors shown in Figure 1(Figure R7) (Figure 1, figure supplement 1).

– We plotted individual mouse data for the defensive behaviors shown in Figure 1 separately for male and female mice (Figure R8) (same as Figure 1, figure supplement 3).

– The fraction of escape-correlated cells increases during exposure to threat compared to control assays (Figure R9)(same as Figure 3I-J).

– PMd-cck photometry signal correlates with escape speed during exposure to rat, but not toy rat (Figure R10).

– We added the chance level for predictions of position and velocity using only PMd-cck ensemble activity. These data show that the prediction error for position and escape velocity are at chance levels during control assays. However, the prediction error is lower than expected by chance for both escape speed and position during exposure to threats (Figure R11) (Same as Figure 4B, 4D and Figure 4B figure supplement 1). Interestingly, PMd-cck activity did not predict approach speed, only escape speed.

– We show CNO treatment does not change approach velocity during the shock grid fear retrieval session (Figure R12).

– We added more animals to show that optogenetic excitation of PMd-cck cells increases locomotion speed (Figure R13).

– We performed simulations showing that non-normal distributions will often pass normality tests using sample sizes typical in behavioral neuroscience (n=12 points). This finding justifies our use of non-parametric tests throughout the manuscript, as standard normality tests are not able to reliably detect deviations from normality unless very large sample sizes are used (Author response image 1).

**Author response image 1. sa2fig1:** (A) Three different distributions were selected to represent a variety of non-normal populations (n=1000). The γ distribution was generated using shape parameter 6 and scale parameter 3. The chi-square distribution plotted has 6 degrees of freedom. The bimodal gaussian distribution has two normal distributions with standard deviation of 1 and means of 6 and 10. (B) Twelve points were selected from each distribution 1000 times, with replacement. Each sample with n=12 points was classified as normal or non-normal by the Lilliefors test for normality. The percent of samples (n=12 points per sample) to pass the Lilliefors normality test is represented by the gray portion of each pie chart. Note that in the vast majority of trials samples selected from non-normal distributions are not classified as significantly non-normal.

– We added a table showing that compared to many typically studied cells, PMd-cck cells show low rheobase and high membrane input resistance, indicating these cells are highly excitable (Table R1).

2) The results presented here must be discussed in the context of other recent findings from the same lab, such as a Neuron paper published by the primary authors of this report (for instance, the Neuron work could provide a rationale for the targeting cck cells that is otherwise missing, and also sheds some light on the negative results obtained for dPM-amv projections).

This issue occurred because this paper was submitted before the publication of the *Neuron* paper, so we couldn’t cite it in the original submission. We now discuss this work in both the Introduction and Discussion sections and use it to justify the rationale of studying PMd cck cells. These prior results are also taken into account when discussing the negative results seen when inhibiting the PMd-amv projection.

Other main results are also discussed in light of this paper. In particular, we outlined the main contributions of this new submission compared to the previous paper. A detailed description of the new results in this paper compared to our previous publication can be seen in Reviewer 1, recommendations for the authors, point 1. We have carefully incorporated the results for the *Neuron* paper in all applicable sections of the current submission.

3) The authors should consider dropping the fMRI results. Though potentially interesting in and of themselves, the relationship between passive viewing of aversive images and escape behavior is not clear. The reviewers felt that these data reduce clarity rather than add insight into what, at best, is a hypothetical homology.

In the original submission we only showed that inhibition of the PMd-dlPAG circuit decreased escape velocity. As pointed out by the Reviewers, this result did not closely parallel the behavioral task used in humans showing increased hypothalamus-dlPAG circuit activity while viewing aversive images. Now we add new data, obtained from mice with dual photometry GCaMP recordings from the PMd and the dlPAG. We now show that mutual information between these two regions increases during exposure to threat compared to the control assays. This analysis was done after excluding all data points with escapes. This new result is thus independent of escape, and is related to exposure to an aversive threat. Thus, these mouse data more closely parallel the result from humans showing higher activity in the hypothalamus-dlPAG pathway during exposure to aversive images, and provides a better rationale for including the human fMRI data.

Importantly, these dual photometry recordings were done contralaterally in each mouse. Consequently, the signal recorded in the dlPAG is created by local cell bodies, and is not contaminated by signals from PMd axons terminating in the dlPAG, as the PMd-cck projection to the dlPAG is unilateral (Figure R2 E-G).

Nevertheless, if the Reviewers still feel strongly that the fMRI data does not fit with the rest of the results, then we can move the fMRI results to the supplemental material, or even remove it altogether. Our preference is to include these results in the main figures of paper, because despite the differences in the human and mouse experiments, the fMRI results are generally well-received and generate interest in audiences when we have presented this work in the past. For example, despite discussing the limitations of this data, Reviewer 1 stated that “Finally, in an intriguing twist, aversive images are shown to increase the functional coupling between hypothalamus and PAG in the human brain. The manuscript is broadly interdisciplinary, spanning multiple subfields of neuroscience research from slice physiology to human brain imaging.”

Reviewer #1 (Recommendations for the authors):A recent publication in Neuron with the same 1st and last authors (Wang et al., 2021, Neuron, 109) seems to substantially scoop this manuscript. Most of the basic finds presented here are conceptual replications of what has been recently demonstrated in Neuron, where the primary authors of this submission to eLife report that dorsal PAG-projecting cck neurons in PMd are activated by and required for escape behavior. The Neuron publication significantly curtails the novelty of this work.

We agree with the Reviewer that the prior paper deals with PMd circuits in the presence of threats. However, none of the main findings in this submission were present in the previous paper. The focus of the *Neuron* paper is on how the PMd controls escape from innate threats in complex contexts that require spatial navigation. There are key results that are not present in the prior publication:

1. The *Neuron* paper has no experiments related to conditioned threats, and in the current paper we show that similar circuit mechanisms control escape from innate and conditioned threats.

2. There is no data on the previous paper about escape vigor, which is the focus in the current paper.

3. Furthermore, the prior paper also does not show that PMd activity can predict escapes in the future, and it does not characterize PMd single cell activity during other behaviors, such as freezing and risk-assessment stretch-attend postures.

4. The prior paper also does not show that PMd single cell activity encodes distance to threat and escape velocity.

5. The *Neuron* paper does not use cfos expression to show how distinct brain regions are activated during PMd optogenetic stimulation, and it does not contain any human fMRI data.

Taken together, these results expand upon the previously published data. Nevertheless, we agree with the Reviewer that both papers deal with related topics, so we significantly altered the discussion to highlight the novel contributions of this paper compared to the *Neuron* publication.

The current manuscript does make a few incremental advances. It delves into the predictive relationship between Ca++ transients in PMd cck neurons and escape to a greater degree than the Neuron paper (though the Neuron paper indeed demonstrates such a relationship).

The prior *Neuron* paper showed that there were cells in the PMd that were active prior to escape. However, this result does not necessarily show that PMd activity can be used to predict escapes in the future. For example, if these patterns of activity didn’t show any consistency across different trials, then PMd activity would not be able to predict future escape. The *Neuron* paper did not demonstrate PMd activity predicts escape in the future, as shown in Figure 3H. Furthermore, in this paper we also demonstrate that PMd single cells are not consistently activated by other defensive behaviors, such as freezing and stretch-attend postures.

We agree with the Reviewer that although different, these results are related, and thus we discuss these new results in light of the previous paper.

It also shows that escape vigor (ie velocity) is mediated by PMd cck circuits (though the fundamental role of this circuitry in escape is established in the Neuron paper). Because of their fundamental similarity, these results submitted to eLife must be discussed in terms of the results published in Neuron so that it is clear to the reader exactly how the current work advances the understanding of escape circuits.

We agree with the Reviewer that it is crucial to clearly explain to the reader how the current results build upon and expand the data from the previous paper. We have thoroughly re-structured the Introduction and the Discussion section to reflect these important changes.

A paper cited by the authors (Cezario et al., 2008) shows a causal role for PMd in defensive reactions to a context previously paired with a predator (in which the predator essentially acts as a US). While this report did not examine escape per se, it does further limit the novelty of what is demonstrated here. It needs to be made clear that a specific role of PMd in quantitative measures of escape is the new result, instead of a broader role role for this region and related circuits in the defensive response to innate and conditioned threats (which has already been established, including by the authors).

We altered the paper to emphasize that the specific role of PMd in quantitative measures of escape is a new result. For example, among other changes, in the end of the abstract we write “Our data identify the PMd-dlPAG circuit as a central node, controlling escape vigor elicited by both innate and conditioned threats.”

The Cezario 2008 paper (Cezario et al., 2008) characterized changes in defensive behavior induced by excitotoxic lesions of the PMd and muscimol infusion in the PMd in rats. Although these results are of significant interest, the methods used have more confounds than those in the current data. The methods used by Cezario do not have high genetic and anatomical specificity, and thus their results may potentially reflect changes caused by actions on nuclei near the PMd, such as the ventromedial hypothalamus.

Our chemogenetic and optogenetic data from this submission and from the prior *Neuron* paper show that inhibiting PMd activity does not affect freezing and stretch-attend postures, while Cezario reports profound impairments in both of these measures following PMd lesions and inactivations with muscimol.

Comparison with our data suggest that some results in Cezario are due to off-target effects on the adjacent ventromedial hypothalamus nucleus, a structure shown both by us (Wang et al., 2021) and others (Wang et al., 2015) to control freezing. Cezario does not provide data on the extent to which muscimol infusions and excitotoxic lesions had off-target effects on other nearby nuclei, such as the posterior and ventromedial nuclei of the hypothalamus, which are structures known to affect defensive behavior. Without such data it is not possible to ascertain which of the reported effects are caused by PMd cells. It is unlikely that the infusion of muscimol and excitotoxins selectively affected only the PMd, which is a very small nucleus surrounded by other nuclei that have distinct roles in defense. Furthermore, the excitotoxic lesions have the additional problem of allowing time for potential compensatory circuit-level changes that further complicate the interpretation of those findings.

In contrast, we use a cck-cre line. Cck is expressed in 92% of PMd cells (Wang et al., 2021), and cck is not expressed in the ventromedial, posterior, dorsomedial or any other nuclei adjacent or near the PMd (Mickelsen et al., 2020), showing that the results reported in this submission cannot be attributed to nuclei other than the PMd. We use genetic targeting and have higher anatomical specificity than lesions and muscimol infusions, thus our data is less susceptible to such off-target effects.

Our data strongly support a role for the PMd specifically in escape but not other defensive behaviors. However, Cezario 2008 did not monitor any escape-related metric. Furthermore, as explained above Cezario’s data may likely have off-target effects. Another potential explanation for this discrepancy is that the PMd may have different functions in rats and mice. Lastly, we provide extensive characterization at the single cell and population-levels of PMd neural activity, while Cezario does not have any comparable datasets. Thus, Cezario’s work does not significantly decrease the novelty of our findings.

It is unclear why the behavioral assays were not counterbalanced. It is possible that recent exposure to predator alters the subjects' response to the aversive stimuli and contextual cues via sensitization or some other process. This weakens the conclusions that can be drawn about the role of premammillary-dPAG projections in behavioral withdrawal from conditioned stimuli associated with shock, thus decreasing the impact of these results.

As explained in Reviewer 1, Public Review, point 2 (Figure R1), we now provide new data showing that prior exposure to the rat assay did not significantly alter any behavioral metric in the shock grid fear retrieval assay. These results have been included in the paper and their significance is discussed as well (Figure 1, figure supplement 2).

The meaning of the fMRI data is unclear. The authors claim that it is not possible to examine the escape behavior in the scanner, but there is a history of avoidance/escape paradigms using fMRI (eg Mobbs et al., 2007, Science, 317). For instance, a simple button press to remove aversive images could serve as an escape response. Absent any such behavioral measure, it is not really possible to understand how hypothalamic-PAG BOLD responses to passive viewing of aversive images relate to the circuit level data obtained in mice. If anything, activity in these regions in the absence of escape behavior seems to belie the central point of the paper.

We agree with the Reviewer that the human and rodent behavioral tasks have important differences, such as lack of escape in the human data.

We now provide new data showing that PMd-dlPAG mutual information increases in the presence of innate and conditioned threats after excluding all time points with escapes (Figure R2 and Figure 9). This result indicates that flow of information in the PMd-dlPAG pathway is higher in the presence of aversive threats, independently of escapes. This new data, which is unrelated to escapes, serves as a better parallel to the fMRI data showing increased activity in the hypothalamic-dlPAG pathway during exposure to aversive visual stimuli. Importantly, dual recordings in PMd and dlPAG cell bodies were done contralaterally to avoid recording signals from GCaMP-expressing PMd axon terminals in the dlPAG.

Reviewer #2 (Recommendations for the authors):While we enjoyed the manuscript overall, we often encountered difficulties in understanding exactly what was done experimentally. We therefore strongly suggest overhauling the methods section to better and unambiguously describe all methodology, with a focus on behavioral analyses. In addition, we would suggest a more careful and comprehensive interpretation of the characterized pathway, leaving room for potential broader functionality. This in our view would take into account the fact that this pathways presents one circuit element within a much larger network, which certainly subserves different functions in a rather dynamic manner.

We thank the Reviewer for these recommendations; we have added detail to the behavioral protocols and analyses, as well as modified our interpretation of the results, allowing room for a broader interpretation of the PMd's functionality. For example, to indicate a broader role of the PMd we added the following text to the Discussion:

“Interestingly, PMd-cck cells also represented distance to threat, but not distance to control stimuli (Figure 4). PMd input to the dlPAG may thus contribute to the encoding of distance to threat and related kinematic variables in dlPAG cells as we recently reported (Reis et al., 2021 and Reis et al., 2021)”

“Our data add to a growing stream of results showing how different components of the medial hypothalamic defense system control threat-induced behaviors, in a densely interconnected network containing the anterior hypothalamus, the ventromedial hypothalamus and the PMd (Cezario et al., 2008). "

We also added detailed timelines for the behavioral assays in and clear definitions of each scored behavior under the Methods section “Behavioral Quantification”

Reviewer #3 (Recommendations for the authors):1. Please provide more specific information in the methods about how the rat exposure and fear conditioning contexts are differentiated. For example, are they made from the same material? Are they cleaned with different solutions?

The rat and fear conditioning contexts are made from different materials. The walls and floor of the rat environment are made of glass. The rat environment is a modified glass aquarium, while the fear conditioning environment is made of laminated white foam board. Both environments are cleaned with ethanol after use. These two assays are also performed in two different lab rooms. This information has been added to the Methods section. These differences produce unique tactile and visual sensory stimuli for each of the boxes.

2. Non-parametric tests are used throughout to determine significance. The rationale behind this decision should be explicitly stated. Were data tested for normality?

We used non-parametric tests throughout because with the sample size used in this and other similar mouse behavioral studies it is often not possible to determine with a reasonable degree of certainty if the distribution of the data is normal.

Author response image 1 shows that random selection of small sample sizes (n=12) from non-normal γ or chi square distributions result in simulated data that passes the Lilliefors normality test over 80% of the time, even though the actual underlying distributions are not normal. For these simulations, a sample size of 12 was used, which is a medium to large-sized sample for a single group in most mouse behavioral papers (such as n=12 mcherry mice, n=12 hm4Di mice).

Thus, we used non-parametric tests to be conservative and avoid making unwarranted assumptions about the underlying distribution of the data with small sample sizes. We now state these justifications in the Methods section, as requested by the Reviewer.

In agreement with our view, prior work in statistics shows that a broad category of major normality tests are extremely underpowered for n<100, which is an unfeasibly large n for rodent cohorts (Razali et al., 2011). The authors compared the power of 4 normality tests (Shapiro-Wilk (SW) test, Kolmogorov-Smirnov (KS) test, Lilliefors (LF) test and Anderson-Darling (AD) test) to detect the normality of samples of different sizes selected from a non-normal Laplace distribution. Normality tests are severely underpowered to detect deviations from normality at sample sizes typically used in behavioral neuroscience (typically n=between 7 and 20 mice per group) (Razali et al., 2011).

3. Terminology for control conditions is inconsistent throughout the figures, e.g. 'control/pre-shock/habituation.'

We removed the term ‘habituation’ as a reference for the control assays. We now use “pre-shock” and “toy rat” for all instances referring to each of the control assays. When referring to both pre-shock and toy-rat assays we used the term “control assays”. We apologize for this inconsistency.

4. The electrophysiological findings are not problematic, but do not strengthen the paper. Without knowing the relative strength of excitatory inputs to the CCK neurons, the excitability measures are not interpretable.

We agree with the Reviewer that it would be informative to characterize the strength of excitatory inputs to PMd-cck cells. However, such experiments are beyond the scope of this paper, in which we only analyze PMd cells and their outputs. Rheobase and membrane input resistance are standard measures commonly used to characterize intrinsic biophysical properties of cells and they provide a measure of how difficult or easy it is to make a cell fire an action potential. These measures are commonly used as properties related to excitability specially in the absence of characterization of inputs.

The rheobase is the minimum amount of current necessary to elicit an action potential. The membrane resistance input is a measure of how ‘leaky’ the cell is. If the cell has low resistance it is ‘leakier’, consequently more current needs to be injected to depolarize the cell. Thus, low rheobase and high input resistance are associated with higher excitability.

Compared to other common cell types, PMd-cck cells have relatively high input resistance and low rheobase, supporting our interpretation that these cells would be able to fire even with relatively weak excitatory inputs. We provide a table for comparison (Author response table 1):

5. The titer of the inhibitory DREADD virus is not reported.

**Author response table 1. sa2table1:** Rheobase and membrane input resistance of PMd and commonly studied cell types. Note that relative to many other cell types PMd cells have relatively low rheobase and high membrane input resistance. Data Source: PMd-cck ((Wang et al., 2021), current report); CA1 (Luque et al., 2017); Striatal medium spiny neurons (Fino et al., 2007); Barrel cortex (Lefort et al., 2009).

	Rheobase (pA)	Input Resistance (MOhms)
PMd cck	38.3± 6.1	484±64
Dorsal Hippocampus CA1	76.64±11.68	151.8±6.90
Striatal medial spiny neurons	155±6	251±11
Barrel cortex (L2 layer)	126±3	188±3
Barrel cortex (L3 layer)	132±4	193±5
Barrel cortex (L4 layer)	56±1	302±4
Barrel cortex (L5a layer)	68±2	210±3
Barrel cortex (L5b layer)	98±3	162±5
Barrel cortex (L6 layer)	76±3	277±4

We apologize for this oversight, and now added that DREADD virus was injected at a titer of 2*10^12^ particles/ml.

6. What is the rationale for the prolonged optogenetic excitation (10 min at 20Hz)?

This 10-minute excitation was used to increase the chance that PMd activation would recruit other regions downstream, inducing cfos expression. Indeed, this protocol was successful and cfos expression increased in several regions following PMd optogenetic activation (Figure 6). However, we did not test other stimulation parameters, and it is possible that shorter epochs of excitation would also produce comparable results.

References

Cezario AF, Ribeiro-Barbosa ER, Baldo MVC, Canteras NS. 2008. Hypothalamic sites responding to predator threats--the role of the dorsal premammillary nucleus in unconditioned and conditioned antipredatory defensive behavior. *Eur J Neurosci* 28:1003–1015.

Chen N, Sugihara H, Kim J, Fu Z, Barak B, Sur M, Feng G, Han W. 2016. Direct modulation of GFAP-expressing glia in the arcuate nucleus bi-directionally regulates feeding. *eLife* 5. doi:10.7554/*eLife*.18716

Fino E, Glowinski J, Venance L. 2007. Effects of acute dopamine depletion on the electrophysiological properties of striatal neurons. *Neurosci Res* 58:305–316.

Halbout B, Marshall AT, Azimi A, Liljeholm M, Mahler SV, Wassum KM, Ostlund SB. 2019. Mesolimbic dopamine projections mediate cue-motivated reward seeking but not reward retrieval in rats. *eLife* 8. doi:10.7554/*eLife*.43551

Ito H, Sales AC, Fry CH, Kanai AJ, Drake MJ, Pickering AE. 2020. Probabilistic, spinally-gated control of bladder pressure and autonomous micturition by Barrington’s nucleus CRH neurons. *eLife*. doi:10.7554/*eLife*.56605

Kwak S, Jung MW. 2019. Distinct roles of striatal direct and indirect pathways in value-based decision making. *eLife* 8. doi:10.7554/*eLife*.46050

Lefort S, Tomm C, Floyd Sarria J-C, Petersen CCH. 2009. The excitatory neuronal network of the C2 barrel column in mouse primary somatosensory cortex. *Neuron* 61:301–316.

Li J, Jiang RY, Arendt KL, Hsu Y-T, Zhai SR, Chen L. 2020. Defective memory engram reactivation underlies impaired fear memory recall in Fragile X syndrome. *eLife* 9. doi:10.7554/*eLife*.61882

Luque MA, Angeles Luque M, Beltran-Matas P, Carmen Marin M, Torres B, Herrero L. 2017. Excitability is increased in hippocampal CA1 pyramidal cells of Fmr1 knockout mice. *PLOS ONE*. doi:10.1371/journal.pone.0185067

Mickelsen LE, Flynn WF, Springer K, Wilson L, Beltrami EJ, Bolisetty M, Robson P, Jackson AC. 2020. Cellular taxonomy and spatial organization of the murine ventral posterior hypothalamus. *eLife* 9. doi:10.7554/*eLife*.58901

Mukherjee D, Gonzales BJ, Ashwal-Fluss R, Turm H, Groysman M, Citri A. 2021. Egr2 induction in spiny projection neurons of the ventrolateral striatum contributes to cocaine place preference in mice. *eLife* 10. doi:10.7554/*eLife*.65228

O’Hare JK, Li H, Kim N, Gaidis E, Ade K, Beck J, Yin H, Calakos N. 2017. Striatal fast-spiking interneurons selectively modulate circuit output and are required for habitual behavior. *eLife*. doi:10.7554/*eLife*.26231

Patel JM, Swanson J, Ung K, Herman A, Hanson E, Ortiz-Guzman J, Selever J, Tong Q, Arenkiel BR. 2019. Sensory perception drives food avoidance through excitatory basal forebrain circuits. *eLife* 8. doi:10.7554/*eLife*.44548

Razali NM, Wah YB, Others. 2011. Power comparisons of shapiro-wilk, kolmogorov-smirnov, lilliefors and anderson-darling tests. *Journal of statistical modeling and analytics*
**2**:21–33.

Wang L, Chen IZ, Lin D. 2015. Collateral pathways from the ventromedial hypothalamus mediate defensive behaviors. *Neuron* 85:1344–1358.

Wang W, Schuette PJ, Nagai J, Tobias BC, Cuccovia V Reis FM, Ji S, de Lima MAX, La-Vu MQ, Maesta-Pereira S, Chakerian M, Leonard SJ, Lin L, Severino AL, Cahill CM, Canteras NS, Khakh BS, Kao JC, Adhikari A. 2021. Coordination of escape and spatial navigation circuits orchestrates versatile flight from threats. *Neuron*. doi:10.1016/j.neuron.2021.03.033